



# Pathfinder v1.0: a Bayesian-inferred simple carbon-climate model to explore climate change scenarios

Thomas Bossy[1,2,*], Thomas Gasser[1,*], and Philippe Ciais[2]

[1]International Institute for Applied Systems Analysis (IIASA), Laxenburg, Austria
[2]Laboratoire des Sciences du Climat et de l'Environnement (LSCE), Gif-sur-Yvette, France
[*]These authors contributed equally to this work.

**Correspondence:** Thomas Gasser (gasser@iiasa.ac.at)

**Abstract.** The Pathfinder model was developed to fill a perceived gap within the range of existing simple climate models. Pathfinder is a compilation of existing formulations describing the climate and carbon cycle systems, chosen for their balance between mathematical simplicity and physical accuracy. The resulting model is simple enough to be used with Bayesian inference algorithms for calibration, which enables assimilation of the latest data from complex Earth system models and the
5 IPCC 6th assessment report, as well as a yearly update based on observations of global temperature and atmospheric CO2. The model's simplicity also enables coupling with integrated assessment models and their optimization algorithms, or running the model in a backward temperature-driven fashion. In spite of this simplicity, the model accurately reproduces behaviours and results from complex models – including uncertainty ranges – when ran following standardized diagnostic experiments. Pathfinder is open-source, and this is its first comprehensive description.

## 1 Introduction

Simple climate models (SCMs) are most often composed of ad hoc parametric laws that emulate the behaviour of more complex Earth system models (ESMs). The emulation allows simulating large ensembles of experiments that would be too costly to compute with ESMs. However, the SCM denomination refers to a fairly broad range of models whose complexity can go from a couple of boxes that only emulate one part of the climate system (e.g. a global temperature impulse response
function; Geoffroy et al., 2013b) to hundreds of state variables representing the different cycles of greenhouse gases and their effect on climate change (e.g. the compact Earth system model OSCAR; Gasser et al., 2017). Simpler models are easier and faster to solve, but they may not adequate for all usages. Therefore, finding the "simplest but not simpler" model depends on a study's precise goals.

In our recent research, we have perceived a deficiency within the existing offer of SCMs, in spite of their large and growing
number (Nicholls et al., 2020). We have therefore developed the Pathfinder model to fill this gap: it is a parsimonious model that carefully balances simplicity and accuracy of representation of physical processes. Pathfinder was designed to fulfil three key requirements: 1. the capacity to be calibrated using Bayesian inference, 2. the capacity to be coupled with integrated assessment models (IAMs), and 3. the capacity to explore a very large number of climate scenarios to narrow down those compatible with limiting climate impacts. The latter motivated the model's name.





While these three requirements clearly call for the simplest model possible, as they all need a fast solving model, they also imply a certain degree of complexity. The Bayesian calibration requires an explicit representation of the processes (i.e. the variables) that are used to constrain the model. Coupling with IAMs requires accurately embedding the latest advances of climate sciences to be policy relevant (National Academies of Sciences and Medicine, 2017). And exploring future climate impacts requires the flexibility to link additional (and potentially regional) impact variables to the core carbon-climate equations.

Here, we present the first public release of Pathfinder. The model is calibrated on Earth system models that contributed to the Coupled Model Intercomparison Project phase 6 (CMIP6), on additional data from the 6th assessment report of the IPCC (AR6), and on observations of global Earth properties up to the year 2021. We first provide a detailed description of the model's equations. We then describe the Bayesian setup used for calibration, the sources of prior information for it, and the resulting posterior configuration. We end with a validation of the model using standard diagnostic simulations and quantitative metrics

for the climate system and carbon cycle.

## 2   Equations

An overview of Pathfinder is presented in Figure 1. The model is composed of a climate module, of three separate modules for the carbon cycle (ocean, land without land use and land permafrost), and of two additional modules describing global impacts: sea level rise (SLR), and surface ocean acidification. We do not emulate cycles of other non-$CO_2$ gases. Mathematically, the

40 model is driven by prescribing time series of any combination of two of four variables: global mean surface temperature (GMST) anomaly (noted $T$), global atmospheric CO2 concentration ($C$), global non-CO2 effective radiative forcing ($R_x$), and global anthropogenic emissions of CO2 ($E_{\mathrm{CO2}}$). The model can therefore be run in the traditional emission-driven and concentration-driven modes, but also in a temperature-driven mode. This is notably important for the calibration, during which it is driven by observations of GMST and atmospheric CO2.

The following presents all equations of the models. Variables are noted using Roman letters, and compiled in Tables A1 and A2. With a few exceptions, parameters are noted using Greek letters, and summarized in Tables A3 and A4. The model has 21 state variables that follow a first-order differential equations in time. The time variable is noted $t$ and kept implicit unless required.

### 2.1   Climate

The GMST change ($T$) induced by effective radiative forcing (ERF; $R$) is represented using a widely used two-box energy balance model with deep ocean heat uptake efficacy (Geoffroy et al., 2013a; Armour, 2017). The first box represents the Earth surface's temperature (including atmosphere, land and surface ocean), and the other one is the deep ocean's temperature ($T_d$). Their time-differential equations are:

$$\Theta_s \frac{\mathrm{d}T}{\mathrm{d}t} = R - \frac{\phi \ln(2)}{T_{2\times}}\, T - \epsilon_{\mathrm{heat}}\, \theta\, (T - T_d) \tag{1}$$



and

$$\Theta_d \frac{\mathrm{d}T_d}{\mathrm{d}t} = \theta \left( T - T_d \right) \tag{2}$$

where $\phi$ is the radiative parameter of CO2, $T_{2\times}$ is the equilibrium climate sensitivity (ECS) at CO2 doubling, $\Theta_s$ is the heat capacity of the surface, $\Theta_d$ is the heat capacity of the deep ocean, $\theta$ is the heat exchange coefficient, and $\epsilon_{\mathrm{heat}}$ is the deep ocean heat uptake efficacy.

The global ERF is simply the sum of the CO2 contribution ($R_{\mathrm{CO2}}$), expressed using the IPCC AR5 formula (Myhre et al., 2013), and that of non-CO2 climate forcers ($R_x$):

$$R = R_{\mathrm{CO2}} + R_x \tag{3}$$

with

$$R_{\mathrm{CO2}} = \phi \ln \left( \frac{C}{C_{\mathrm{pi}}} \right) \tag{4}$$

where $C_{\mathrm{pi}}$ is the preindustrial atmospheric CO2 concentration.

The above energy balance model naturally provides the ocean heat content (OHC; $U_{\mathrm{ohc}}$) as:

$$U_{\mathrm{ohc}} = \alpha_{\mathrm{ohc}} \left( \Theta_s \, T + \Theta_d \, T_d \right) \tag{5}$$

and the ocean heat uptake (OHU) as:

$$\frac{\mathrm{d}U_{\mathrm{ohc}}}{\mathrm{d}t} = \alpha_{\mathrm{ohc}} \left( \Theta_s \, \frac{\mathrm{d}T}{\mathrm{d}t} + \Theta_d \, \frac{\mathrm{d}T_d}{\mathrm{d}t} \right) \tag{6}$$

where $\alpha_{\mathrm{ohc}}$ is the fraction of energy used to warm the ocean (i.e. excluding the energy needed to heat up the atmosphere and land, and to melt ice).

## 2.2 Sea level rise

Global SLR has been implemented in Pathfinder as a variable of interest to model climate change impacts. In this version, it is firstly a proof of concept, modelled in a simple yet sensible manner. The total sea level rise ($H_{\mathrm{tot}}$) is the sum of contributions from thermal expansion ($H_{\mathrm{thx}}$), Greenland ice sheet (GIS; $H_{\mathrm{gis}}$), Antarctica ice sheet (AIS; $H_{\mathrm{ais}}$), and glaciers ($H_{\mathrm{gla}}$):

$$H_{\mathrm{tot}} = H_{\mathrm{thx}} + H_{\mathrm{gis}} + H_{\mathrm{ais}} + H_{\mathrm{gla}} \tag{7}$$

The thermal expansion contribution scales linearly with the OHC (Goodwin et al., 2017; Fox-Kemper et al., 2021):

$$H_{\mathrm{thx}} = \Lambda_{\mathrm{thx}} \, U_{\mathrm{ohc}} \tag{8}$$

where $\Lambda_{\mathrm{thx}}$ is the scaling factor of the thermosteric contribution to SLR. Note, however, that the thermal capacity of the climate module does not match that of the real-world ocean (Geoffroy et al., 2013b), and so this equation cannot describe equilibrium SLR over millennial timescales.





To model contributions from ice sheets and glaciers, we followed the general approach of Mengel et al. (2016). The SLR caused by GIS follows a first-order differential equation with its specific timescale, and the equilibrium SLR from GIS is assumed to be a cubic function of GMST:

$$\frac{\mathrm{d}H_{\mathrm{gis}}}{\mathrm{d}t} = \lambda_{\mathrm{gis}} + \frac{1}{\tau_{\mathrm{gis}}} \left( \Lambda_{\mathrm{gis1}}\, T + \Lambda_{\mathrm{gis3}}\, T^3 - H_{\mathrm{gis}} \right) \tag{9}$$

where $\lambda_{\mathrm{gis}}$ is an offset parameter introduced because GIS was not in a steady state at the end of the preindustrial era, $\Lambda_{\mathrm{gis1}}$ is the linear term of equilibrium of GIS SLR, $\Lambda_{\mathrm{gis3}}$ is the cubic term of equilibrium of GIS SLR, and $\tau_{\mathrm{gis}}$ is the timescale of the GIS contribution. The motivation for replacing the quadratic term of Mengel et al. (2016) by a cubic one is the oddness of the cubic function that leads to negative (and not positive) SLR for negative $T$ (which happens during the earlier years of the calibration run).

The contribution from glaciers is also a first-order differential equation with an equilibrium inspired by Mengel et al. (2016). We expanded it with a cubic term to account for the fact that we aggregate all glaciers together and allow more skewness in the curve describing the equilibrium SLR as a function of $T$. In addition, we added an exponential sensitivity to speed up the convergence to equilibrium under warmer climate:

$$\frac{\mathrm{d}H_{\mathrm{gla}}}{\mathrm{d}t} = \lambda_{\mathrm{gla}} + \frac{\exp(\gamma_{\mathrm{gla}}\, T)}{\tau_{\mathrm{gla}}} \left( \Lambda_{\mathrm{gla}} \left( 1 - \exp\!\left( -\Gamma_{\mathrm{gla1}}\, T - \Gamma_{\mathrm{gla3}}\, T^3 \right) \right) - H_{\mathrm{gla}} \right) \tag{10}$$

where $\lambda_{\mathrm{gla}}$ is an offset parameter accounting for the lack of initial steady-state, $\Lambda_{\mathrm{gla}}$ is the SLR potential if all glaciers melted, $\Gamma_{\mathrm{gla1}}$ is the linear sensitivity of glaciers' equilibrium to climate change, $\Gamma_{\mathrm{gla3}}$ is the cubic sensitivity of glaciers' equilibrium to climate change, $\tau_{\mathrm{gla}}$ is the timescale of the glaciers contribution, and $\gamma_{\mathrm{gla}}$ is the sensitivity of glaciers' timescale to climate change.

Following Mengel et al. (2016), the contribution from AIS is further divided in two terms, one for surface mass balance (SMB; $H_{\mathrm{ais,smb}}$) and one for solid ice discharge (SID; $H_{\mathrm{ais,sid}}$), so that $H_{\mathrm{ais}} = H_{\mathrm{ais,smb}} + H_{\mathrm{ais,sid}}$. It is expected that precipitation will increase over Antarctica under higher GMST, leading to increase in SMB and to a negative sea level rise contribution modeled as:

$$\frac{\mathrm{d}H_{\mathrm{ais,smb}}}{\mathrm{d}t} = -\Lambda_{\mathrm{ais,smb}}\, T \tag{11}$$

where $\Lambda_{\mathrm{ais,smb}}$ is the AIS SMB sensitivity to climate change (expressed in sea level equivalent). At the same time, increasing surface ocean temperatures will cause more SID through basal melting, which we model using a first-order differential equation assumed to be independent of the SMB effect, and with a term that speeds up the effect the more SID happened:

$$\frac{\mathrm{d}H_{\mathrm{ais,sid}}}{\mathrm{d}t} = \lambda_{\mathrm{ais}} + \frac{1 + \alpha_{\mathrm{ais}}\, H_{\mathrm{ais,sid}}}{\tau_{\mathrm{ais}}} \left( \Lambda_{\mathrm{ais}}\, T - H_{\mathrm{ais,sid}} \right) \tag{12}$$

where $\lambda_{\mathrm{ais}}$ is an offset parameter accounting for the lack of initial steady-state, $\Lambda_{\mathrm{ais}}$ is the SLR equilibrium of AIS SID, $\tau_{\mathrm{ais}}$ is the timescale of the AIS SID contribution, and $\alpha_{\mathrm{ais}}$ is the sensitivity of the timescale to past SID. In the model's code, however, we directly solve for the total AIS contribution as:

$$\frac{\mathrm{d}H_{\mathrm{ais}}}{\mathrm{d}t} = -\Lambda_{\mathrm{ais,smb}}\, T + \lambda_{\mathrm{ais}} + \frac{1 + \alpha_{\mathrm{ais}}\, (H_{\mathrm{ais}} - H_{\mathrm{ais,smb}})}{\tau_{\mathrm{ais}}} \left( \Lambda_{\mathrm{ais}}\, T - (H_{\mathrm{ais}} - H_{\mathrm{ais,smb}}) \right) \tag{13}$$



## 2.3 Ocean carbon

To calculate the ocean carbon sink, we use the classic mixed-layer impulse response function model from Joos et al. (1996),
updated to the equivalent box-model formulation of Strassmann and Joos (2018), and extended in places to introduce parameter
adjustments for calibration. In the model, the mixed layer is split into 5 boxes (subscript $j$), as represented in Figure 2, so that
the total carbon in the mixed layer pool ($C_o$) is:

$$C_o = \sum_j C_{o,j} \tag{14}$$

This total carbon mass is converted into a molar concentration of dissolved inorganic carbon (DIC; $c_{\mathrm{dic}}$) following:

$$c_{\mathrm{dic}} = \frac{\alpha_{\mathrm{dic}}}{\beta_{\mathrm{dic}}} C_o \tag{15}$$

where $\alpha_{\mathrm{dic}}$ is a fixed conversion factor, and $\beta_{\mathrm{dic}}$ is a scaling factor for the conversion. (The latter can be seen as a factor
multiplying the mixed layer depth: it is 1 if the depth is unchanged from the original Strassmann and Joos (2018) model.)

The non-linear carbonate chemistry in the mixed layer is emulated in two steps. First, the model's original polynomial
function is used to determine the partial pressure of CO2 affected by changes in DIC only ($p_{\mathrm{dic}}$):

$$
\begin{aligned}
p_{\mathrm{dic}} = {} & (1.5568 - 0.013993\, T_o)\, c_{\mathrm{dic}} \\
& + (7.4706 - 0.20207\, T_o)\, 10^{-3}\, c_{\mathrm{dic}}{}^2 \\
& - (1.2748 - 0.12015\, T_o)\, 10^{-5}\, c_{\mathrm{dic}}{}^3 \\
& + (2.4491 - 0.12639\, T_o)\, 10^{-7}\, c_{\mathrm{dic}}{}^4 \\
& - (1.5768 - 0.15326\, T_o)\, 10^{-10}\, c_{\mathrm{dic}}{}^5
\end{aligned}
\tag{16}
$$

where $T_o$ is the preindustrial surface ocean temperature. Second, the actual partial pressure of CO2 ($p_{\mathrm{CO2}}$) is calculated using
an exponential climate sensitivity (Takahashi et al., 1993; Joos et al., 2001):

$$p_{\mathrm{CO2}} = (p_{\mathrm{dic}} + C_{\mathrm{pi}})\, \exp(\gamma_{\mathrm{dic}}\, T) \tag{17}$$

where $\gamma_{\mathrm{dic}}$ is the sensitivity of $p_{\mathrm{CO2}}$ to climate change.

The flux of carbon between the atmosphere and the ocean ($F_{\mathrm{ocean}}$, defined positively if it is a carbon sink) is caused by
the difference in partial pressure of CO2 in the atmosphere and at the oceanic surface, following an exchange rate that varies
linearly with GMST, that is here used as a proxy for wind changes:

$$F_{\mathrm{ocean}} = \nu_{\mathrm{gx}}\, (1 + \gamma_{\mathrm{gx}}\, T)\, (C - p_{\mathrm{CO2}}) \tag{18}$$

where $\nu_{\mathrm{gx}}$ is the preindustrial gas-exchange rate, and $\gamma_{\mathrm{gx}}$ is its sensitivity to climate change.

This flux of carbon entering the ocean is split between the mixed layer carbon subpools, and this added carbon is subsequently
transported towards the deep ocean at a rate specific to each subpool. This leads to the following differential equations:

$$\frac{\mathrm{d}C_{o,j}}{\mathrm{d}t} = -\frac{C_{o,j}}{\kappa_{\tau_o}\, \tau_{o,j}} + \alpha_{o,j}\, F_{\mathrm{ocean}}, \quad \forall j \tag{19}$$




where $\alpha_{o,j}$ are the subpools' splitting shares (with $\sum_j \alpha_{o,j} = 1$), $\tau_{o,j}$ are the subpools' timescales for transport to the deep ocean, and $\kappa_{\tau_o}$ is a scaling factor applied to all subpools. Finally, the deep ocean carbon pool ($C_d$) is obtained through mass balance:

$$\frac{\mathrm{d}C_d}{\mathrm{d}t} = \sum_j \frac{C_{o,j}}{\kappa_{\tau_o}\,\tau_{o,j}} \tag{20}$$

### 2.4 Ocean acidification

While in the real world, ocean acidification is directly related to the carbonate chemistry and the ocean sink, we do not have a simple formulation at our disposal that could link it to our ocean carbon cycle module. We therefore use a readily available emulation of the surface ocean acidification ($\mathrm{pH}$) that links it directly to the atmospheric concentration of CO2 (Bernie et al., 2010) with the following polynomial approximation:

$$\mathrm{pH} = \kappa_{\mathrm{pH}} \left(8.5541 - 0.00173\,C + 1.3264\,10^{-6}\,C^2 - 4.4943\,10^{-10}\,C^3\right) \tag{21}$$

where $\kappa_{\mathrm{pH}}$ is a scaling factor (that defaults to 1). We note that this approach is reasonable for the surface ocean, as it quickly equilibrates with the atmosphere (but it would not work for the deep ocean).

### 2.5 Land carbon

The land carbon module of Pathfinder is a simplified version of the one in OSCAR (Gasser et al., 2017, 2020). It is shrunk down to four global carbon pools: vegetation, litter, active and passive soil (see Figure 3). All terrestrial biomes are lumped together, and there is therefore no accounting of the impact of land use change on the land carbon cycle in this version of Pathfinder. This is an extreme assumption – although very common in SCMs – motivated by simplicity, and it implies that CO2 emissions from fossil fuel burning and land use change are assumed to behave in the exact same way, in spite of their not doing so in reality (Gitz and Ciais, 2003; Gasser and Ciais, 2013).

The vegetation carbon pool ($C_v$) results from the balance between net primary productivity (NPP; $F_{\mathrm{npp}}$), emission from wildfires ($E_{\mathrm{fire}}$), emission from harvest and grazing ($E_{\mathrm{harv}}$), and loss of carbon from biomass mortality ($F_{\mathrm{mort}}$):

$$\frac{\mathrm{d}C_v}{\mathrm{d}t} = F_{\mathrm{npp}} - E_{\mathrm{fire}} - E_{\mathrm{harv}} - F_{\mathrm{mort}} \tag{22}$$

NPP is expressed as its own preindustrial value multiplied by a function of CO2 and of GMST ($r_{\mathrm{npp}}$). This function thus embeds the so-called CO2-fertilisation effect, described using a generalised logarithmic functional form:

$$F_{\mathrm{npp}} = F_{\mathrm{npp0}}\,r_{\mathrm{npp}} \tag{23}$$

with

$$r_{\mathrm{npp}} = \left(1 + \frac{\beta_{\mathrm{npp}}}{\alpha_{\mathrm{npp}}}\left(1 - \left(\frac{C}{C_{\mathrm{pi}}}\right)^{-\alpha_{\mathrm{npp}}}\right)\right)(1 + \gamma_{\mathrm{npp}}\,T) \tag{24}$$



where $F_{\mathrm{npp0}}$ is the preindustrial NPP, $\beta_{\mathrm{npp}}$ is the CO2-fertilisation sensitivity, $\alpha_{\mathrm{npp}}$ is the CO2-fertilisation shape parameter for saturation (with $r_{\mathrm{npp}} \to (1+\beta_{\mathrm{npp}}\ln(C/C_{\mathrm{pi}}))(1+\gamma_{\mathrm{npp}}T)$ as $\alpha_{\mathrm{npp}} \to 0^+$), and $\gamma_{\mathrm{npp}}$ is the sensitivity of NPP to climate change.

Harvesting and mortality fluxes are taken proportional to the carbon pool itself even though in reality the mortality fluxes
are climate dependent. For simplicity we assume a constant mortality following the equations in OSCAR (Gasser et al., 2017):

$$E_{\mathrm{harv}} = \nu_{\mathrm{harv}}\, C_v \qquad (25)$$

and

$$F_{\mathrm{mort}} = \nu_{\mathrm{mort}}\, C_v \qquad (26)$$

where $\nu_{\mathrm{harv}}$ is the harvesting/grazing rate, and $\nu_{\mathrm{mort}}$ is the mortality rate.

Wildfires emissions are also assumed proportional to the vegetation carbon pool, but with an additional linear dependency of the emission rate on CO2 and GMST ($r_{\mathrm{fire}}$):

$$E_{\mathrm{fire}} = \nu_{\mathrm{fire}}\, r_{\mathrm{fire}}\, C_v \qquad (27)$$

with

180 $$r_{\mathrm{fire}} = \left(1 + \beta_{\mathrm{fire}}\left(\frac{C}{C_{\mathrm{pi}}} - 1\right)\right)(1 + \gamma_{\mathrm{fire}}\, T) \qquad (28)$$

where $\nu_{\mathrm{fire}}$ is the wildfires rate, $\beta_{\mathrm{fire}}$ is the sensitivity of wildfires to CO2, and $\gamma_{\mathrm{fire}}$ is their sensitivity to climate change.

Soil carbon is divided into three pools. The litter carbon pool ($C_{s1}$) receives the mortality flux as sole input, it emits part of its carbon through heterotrophic respiration ($E_{\mathrm{rh1}}$), and it transfers another part to the next pool through stabilization ($F_{\mathrm{stab}}$):

$$\frac{\mathrm{d}C_{s1}}{\mathrm{d}t} = F_{\mathrm{mort}} - F_{\mathrm{stab}} - E_{\mathrm{rh1}} \qquad (29)$$

Similarly, the active soil carbon pool ($C_{s2}$) receives the stabilization flux, is respired ($E_{\mathrm{rh2}}$), and transfers carbon to the last pool through passivization ($F_{\mathrm{pass}}$):

$$\frac{\mathrm{d}C_{s2}}{\mathrm{d}t} = F_{\mathrm{stab}} - F_{\mathrm{pass}} - E_{\mathrm{rh2}} \qquad (30)$$

The passive carbon pool ($C_{s3}$) receives this final input flux and is respired ($E_{\mathrm{rh3}}$):

$$\frac{\mathrm{d}C_{s3}}{\mathrm{d}t} = F_{\mathrm{pass}} - E_{\mathrm{rh3}} \qquad (31)$$

Although information pertaining to this fourth pool is not commonly provided by ESMs, it was introduced in Pathfinder to adjust the complex models' turnover time of soil carbon to better match isotopic data (He et al., 2016). For completeness, we note that the total heterotrophic respiration is $E_{\mathrm{rh}} = E_{\mathrm{rh1}} + E_{\mathrm{rh2}} + E_{\mathrm{rh3}}$, and the total soil carbon pool is $C_s = C_{s1} + C_{s2} + C_{s3}$.





All soil-originating fluxes are taken proportional to their pool of origin, and multiplied by a function ($r_{\mathrm{rh}}$) explained hereafter. For the litter pool, this gives:

$$E_{\mathrm{rh1}} = \nu_{\mathrm{rh1}}\, r_{\mathrm{rh}}\, C_{s1} \tag{32}$$

and

$$F_{\mathrm{stab}} = \nu_{\mathrm{stab}}\, r_{\mathrm{rh}}\, C_{s1} \tag{33}$$

where $\nu_{\mathrm{rh1}}$ is the litter respiration rate, and $\nu_{\mathrm{stab}}$ is the stabilization rate. For the active soil pool, we have:

$$E_{\mathrm{rh2}} = \frac{\nu_{\mathrm{rh23}} - \nu_{\mathrm{rh3}}\, \alpha_{\mathrm{pass}}}{1 - \alpha_{\mathrm{pass}}}\, r_{\mathrm{rh}}\, C_{s2} \tag{34}$$

and

$$F_{\mathrm{pass}} = \nu_{\mathrm{rh3}}\, \frac{\alpha_{\mathrm{pass}}}{1 - \alpha_{\mathrm{pass}}}\, r_{\mathrm{rh}}\, C_{s2} \tag{35}$$

and for the passive soil pool:

$$E_{\mathrm{rh3}} = \nu_{\mathrm{rh3}}\, r_{\mathrm{rh}}\, C_{s3} \tag{36}$$

where $\nu_{\mathrm{rh23}}$ is the soil respiration rate (averaged over active and passive pools), $\nu_{\mathrm{rh3}}$ is the passive soil respiration rate, and $\alpha_{\mathrm{pass}}$ is the fraction of passive carbon (over active+passive soil carbon). This slightly convoluted formulation is motivated by the lack of information regarding the active/passive split in ESMs, which we alleviate using additional data during calibration.

In addition, the function $r_{\mathrm{rh}}$, describing the dependency of respiration (and related fluxes) on temperature and on the availability of fresh organic matter to be decomposed, is defined as:

$$r_{\mathrm{rh}} = \frac{\underbrace{\left(1 + \beta_{\mathrm{rh}}\left(\frac{C_{s1}}{C_{s1} + C_{s2} + C_{s3}}\left(1 + \frac{\nu_{\mathrm{stab}}}{\nu_{\mathrm{rh23}}}\right) - 1\right)\right)}\exp(\gamma_{\mathrm{rh}}\, T)}{\left(1 + \beta_{\mathrm{rh}}\left(\frac{C_{s1}}{C_s}\frac{C_s(t_0)}{C_{s1}(t_0)} - 1\right)\right)} \tag{37}$$

where $\beta_{\mathrm{rh}}$ is the sensitivity of the respiration to fresh organic matter availability (expressed here as the relative change in the $C_{s1}/C_s$ ratio with regard to preindustrial times), and $\gamma_{\mathrm{rh}}$ is its sensitivity to climate change (equivalent to a "$Q_{10}$" formulation with $Q_{10} = \exp(10\,\gamma_{\mathrm{rh}})$).

Finally, the net carbon flux from the atmosphere to the land ($F_{\mathrm{land}}$, defined positively if it is a carbon sink) is obtained as the net budget of all pools combined:

$$F_{\mathrm{land}} = F_{\mathrm{npp}} - E_{\mathrm{fire}} - E_{\mathrm{harv}} - E_{\mathrm{rh}} \tag{38}$$

and this system of equations leads to the following preindustrial steady-state:

$$\begin{cases} C_v(t_0) = \frac{F_{\mathrm{npp0}}}{\nu_{\mathrm{fire}} + \nu_{\mathrm{harv}} + \nu_{\mathrm{mort}}} \\ C_{s1}(t_0) = C_v(t_0)\frac{\nu_{\mathrm{mort}}}{\nu_{\mathrm{rh1}} + \nu_{\mathrm{stab}}} \\ C_{s2}(t_0) = C_{s1}(t_0)\frac{\nu_{\mathrm{stab}}}{\nu_{\mathrm{rh23}}}(1 - \alpha_{\mathrm{pass}}) \\ C_{s3}(t_0) = C_{s1}(t_0)\frac{\nu_{\mathrm{stab}}}{\nu_{\mathrm{rh23}}}\alpha_{\mathrm{pass}} \end{cases} \tag{39}$$



## 2.6 Permafrost carbon

As the land carbon cycle described in the previous section does not account for permafrost carbon, we implemented this
feedback using the emulator developed by Gasser et al. (2018) but aggregated into a unique global region. The emulation starts
with a theoretical thawed fraction ($\bar{a}$) that represents the fraction of thawed carbon under steady-state for a certain level of local
warming. It is formulated with a sigmoid function (that equals 0 at preindustrial and 1 under very high GMST):

$$\bar{a} = -a_{\min} + \frac{(1 + a_{\min})}{\left(1 + \left(\left(1 + \frac{1}{a_{\min}}\right)^{\kappa_a} - 1\right)\exp(-\gamma_a\,\kappa_a\,\alpha_{\mathrm{lst}}\,T)\right)^{\frac{1}{\kappa_a}}} \tag{40}$$

where $-a_{\min}$ is the minimum thawed fraction (corresponding to 100% frozen soil carbon), $\kappa_a$ is a shape parameter determining
the asymmetry of the function, $\gamma_a$ is the sensitivity of the theoretical thawed fraction to local climate change, and $\alpha_{\mathrm{lst}}$ is the
proportionality factor between local and global climate change.

The actual thawed fraction ($a$) then moves towards its theoretical value at a speed that depends on whether it is thawing (i.e.
$a < \bar{a}$) or freezing (i.e. $a > \bar{a}$). This is written as a non-linear differential equation:

$$\frac{\mathrm{d}a}{\mathrm{d}t} = 0.5\,(\nu_{\mathrm{thaw}} + \nu_{\mathrm{froz}})\,(\bar{a} - a) + 0.5\,|(\nu_{\mathrm{thaw}} - \nu_{\mathrm{froz}})\,(\bar{a} - a)| \tag{41}$$

where $\nu_{\mathrm{thaw}}$ is the rate of thawing, and $\nu_{\mathrm{froz}}$ is the rate of freezing. Because $\nu_{\mathrm{thaw}} > \nu_{\mathrm{froz}}$, the absolute value in the equation
leads to the right-hand side being $\nu_{\mathrm{thaw}}(\bar{a} - a)$ if $a < \bar{a}$, or $\nu_{\mathrm{froz}}(\bar{a} - a)$ if $a > \bar{a}$. The change in the pool of frozen carbon ($C_{\mathrm{fr}}$)
naturally follows:

$$\frac{\mathrm{d}C_{\mathrm{fr}}}{\mathrm{d}t} = -\frac{\mathrm{d}a}{\mathrm{d}t}\,C_{\mathrm{fr}0} \tag{42}$$

where $C_{\mathrm{fr}0}$ is the amount of frozen carbon at preindustrial times.

Thawed carbon is not directly emitted to the atmosphere: it is split into three thawed carbon subpools ($C_{\mathrm{th},j}$) that have their
own decay time, but are all affected by an additional function ($r_{\mathrm{rt}}$). This leads to the following budget equations:

$$\frac{\mathrm{d}C_{\mathrm{th},j}}{\mathrm{d}t} = -\alpha_{\mathrm{th},j}\,\frac{\mathrm{d}C_{\mathrm{fr}}}{\mathrm{d}t} - \frac{C_{\mathrm{th},j}}{\kappa_{\tau_{\mathrm{th}}}\,\tau_{\mathrm{th},j}}\,r_{\mathrm{rt}}, \quad \forall j \tag{43}$$

where $\alpha_{\mathrm{th},j}$ are the subpools' splitting shares (with $\sum_j \alpha_{\mathrm{th},j} = 1$), $\tau_{\mathrm{th},j}$ are the subpools' decay times, and $\kappa_{\tau_{\mathrm{th}}}$ is a scaling
factor applied to all subpools. The additional $r_{\mathrm{rt}}$ function describes the sensitivity of heterotrophic respiration to climate change
in boreal regions, using a Gaussian formula:

$$r_{\mathrm{rt}} = \exp\left(\kappa_{\mathrm{rt}}\,\gamma_{\mathrm{rt}1}\,\alpha_{\mathrm{lst}}\,T - \kappa_{\mathrm{rt}}\,\gamma_{\mathrm{rt}2}\,(\alpha_{\mathrm{lst}}\,T)^2\right) \tag{44}$$

where $\kappa_{\mathrm{rt}}$ is a factor scaling the sensitivity of thawed carbon against that of regular soil carbon, $\gamma_{\mathrm{rt}1}$ is the sensitivity to local
temperature change (i.e. a Q10), and $\gamma_{\mathrm{rt}2}$ is the quadratic term in the latter sensitivity that represents a saturation effect. Noting
that all the emitted carbon is assumed to be CO2, the global emission from permafrost ($E_{\mathrm{pf}}$) is thus:

$$E_{\mathrm{pf}} = \sum_j \frac{C_{\mathrm{th},j}}{\kappa_{\tau_{\mathrm{th}}}\,\tau_{\mathrm{th},j}}\,r_{\mathrm{rt}} \tag{45}$$





## 2.7 Atmospheric CO2

The change in atmospheric concentration of CO2 is the budget of all carbon cycle fluxes to which we add the exogenous anthropogenic emissions ($E_{CO2}$):

$$\alpha_C \frac{dC}{dt} = E_{CO2} + E_{pf} - F_{land} - F_{ocean} \tag{46}$$

where $\alpha_C$ is the conversion factor from volume fraction to mass for CO2.

## 3  Bayesian calibration

### 3.1  Principle and setup

Bayesian inference is a powerful tool for assimilating observational data into reduced-complexity models such as Pathfinder (Ricciuto et al., 2008). The approach consists in deducing joint probability distributions of parameters from a priori knowledge

on those distributions and on distributions of observations of some of the model's state variables, using Bayes' theorem (Bayes, 1763).

Concretely, the posterior probability $\mathcal{P}_{post}$ of a sample $k$ from the joint parameters distribution $\boldsymbol{\xi}_k$, conditional to a set of observations $\boldsymbol{x}$, is proportional (symbol $\propto$) to its own prior probability $\mathcal{P}_{pre}$ and to the likelihood $\mathcal{L}$ of the model simulating $\boldsymbol{x}$ given $\boldsymbol{\xi}_k$:

$$\mathcal{P}_{post}(\boldsymbol{\xi}_k|\boldsymbol{x}) \propto \mathcal{L}(\boldsymbol{x}|\boldsymbol{\xi}_k)\, \mathcal{P}_{pre}(\boldsymbol{\xi}_k) \tag{47}$$

Here, we assume all observations are independently and identically distributed following a normal distribution (with mean values $\boldsymbol{\mu_x}$, and standard deviations $\boldsymbol{\sigma_x}$ expressed in real physical units), which leads to the following likelihood:

$$\mathcal{L}(\boldsymbol{x}|\boldsymbol{\xi}_k) = \prod_{i=1}^{n_x} \frac{1}{\sigma_{x,i}\sqrt{2\pi}} \exp\left(-\frac{(\mathcal{F}_i(\boldsymbol{\xi}_k) - \mu_{x,i})^2}{2\,\sigma_{x,i}^2}\right) \tag{48}$$

where $\mathcal{F}_i(\boldsymbol{\xi}_k)$ is the model's output for the $i$-th observable (out of $n_x$) with input parameters $\boldsymbol{\xi}_k$.

The Pathfinder model is a set of (discretized) differential equations with a number of input parameters, of which $n_\xi$ are calibrated through Bayesian inference, and an additional two input variables provided as time series (i.e. one value per time step required). While the two input time series can be any combination of two out of four variables (anthropogenic CO2 emissions, non-CO2 ERF, atmospheric CO2 concentration, or GMST), for calibration we use the two most well constrained variables that are direct physical observations of the global Earth system: atmospheric CO2 and GMST. These input time series

cover the historical period from 1751 to 2020. Therefore, the $\boldsymbol{\xi}_k$ vector is:

$$\boldsymbol{\xi}_k = \left\{\{\xi_j\}_{j=1}^{n_\xi}, \{C(t)\}_{t=1751}^{2020}, \{T(t)\}_{t=1751}^{2020}\right\}_k \tag{49}$$

However, to ease the computation by reducing the dimension of the system, we do not use annual time series of observations as inputs, but we assume that each input time series (for variable $X$ being $C$ or $T$) follows:

$$X(t) = X_\mu(t) + \tilde{\sigma}_X\, X_\sigma(t) + \epsilon_X\, \text{AR1}(t;\rho_X) \tag{50}$$





where $X_\mu$ and $X_\sigma$ are fixed exogenous annual time series (i.e. structural parameters), $\tilde{\sigma}_X$ is the relative standard deviation of the time series (without noise), $\epsilon_X$ is the noise intensity, and AR1 is an autoregressive process of order 1 and autocorrelation parameter $\rho_X$. This assumption leads to the final expression of the $\boldsymbol{\xi}_k$ vector:

$$\boldsymbol{\xi}_k = \left\{ \left\{ \xi_j \right\}_{j=1}^{n_\xi}, \tilde{\sigma}_C, \epsilon_C, \rho_C, \tilde{\sigma}_T, \epsilon_T, \rho_T, \right\}_k \qquad (51)$$

The Bayesian procedure is implemented using the Python computer language, specifically the PyMC3 package (Salvatier
et al., 2016). The distribution sampling and parameter distributions estimation are done through a full-rank Automatic Differentiation Variational Inference (ADVI) algorithm (Kucukelbir et al., 2017), with 100,000 iterations (and default algorithm options). A strength of full-rank ADVI is the algorithm's speed, and its ability to generate correlated (i.e. joint) posterior distributions even if the prior ones are uncorrelated. For the computation, the differential system of Pathfinder is solved using an implicit-explicit scheme, with a time step of one quarter of a year.

### 3.2  Parameters

Out of the model's 77 parameters, 33 are assumed to be fixed (i.e. they are structural parameters), and the remaining $n_\xi$ = 44 parameters are estimated through Bayesian inference. Prior distributions of the $\xi_j$ parameters are assumed log-normal if the physical parameter must be defined positive, logit-normal if it must be between 0 and 1, and normal otherwise. To avoid extreme parameter values that could make the model diverge during calibration, the posterior distributions are bound to
$\mu_{\xi,j} \pm 5\sigma_{\xi,j}$, where $\mu_{\xi,j}$ and $\sigma_{\xi,j}$ are the mean and standard deviation of the $j$-th parameter's prior distribution. These two values are taken from the literature, deduced from multi-model ensembles, or in a few instances arbitrarily set, as described in the following subsections. Note that when parameters are deduced from multi-model ensembles, there are effectively two rounds of calibration: first, a calibration on individual models using ordinary least square regressions to obtain prior distributions, and second, the Bayesian calibration itself that leads to the posterior distributions. In addition, the prior distributions of $\tilde{\sigma}_X$, $\epsilon_X$ and
$\rho_X$ are assumed normal, half-normal and uniform, respectively (see Figure 5). All prior distributions are assumed independent, so that the prior joint distribution $\boldsymbol{\xi}$ does not exhibit any covariance. All parameters are summarised in Tables A5 and **??** along with their properties and values.

#### 3.2.1  Climate

All the parameters of the climate module are calibrated. The prior radiative parameter $\phi$ is taken from the AR5 (Myhre et al.,
2013, Table 8.SM.1). All other prior parameters of the climate module (i.e. $T_{2\times}$, $\Theta_s$, $\Theta_d$, $\theta$ and $\epsilon_{\text{heat}}$) are taken from 35 CMIP6 models whose climate responses were derived for the AR6 using the *abrupt-4xCO2* experiment (Smith et al., 2021, Section 7.SM.2.1, and corresponding GitHub repository). Here, $T_{2\times}$ is simply assumed to be half the reported equilibrium temperature at quadrupled CO2. In addition, the prior ocean warming fraction $\alpha_{\text{ohc}}$ is taken from the AR6 (Forster et al., 2021, Table 7.1).





### 3.2.2 Sea level rise

Some parameters from the SLR module are structural: the maximum SLR contribution from glaciers ($\Lambda_{\mathrm{gla}}$) is taken from Fox-Kemper et al. (2021, Section 9.6.3.2), the equilibrium AIS SLR ($\Lambda_{\mathrm{ais}}$) is from (Church et al., 2013, Figure 13.14), and the $\tau_{\mathrm{gis}}$, $\tau_{\mathrm{gla}}$ and $\tau_{\mathrm{ais}}$ timescales are the mean values from Mengel et al. (2016, Table S1) (assuming they provide the $90\%$-range of a log-normal distribution). All other parameters are calibrated. The prior thermosteric parameter $\Lambda_{\mathrm{thx}}$ is taken from the AR6 (Fox-Kemper et al., 2021, Section 9.2.4.1), as are the prior preindustrial offset parameters $\lambda_{\mathrm{gis}}$, $\lambda_{\mathrm{gla}}$ and $\lambda_{\mathrm{ais}}$ (Fox-Kemper 310 et al., 2021, earliest period of Table 9.5). For the remaining parameters, we derive prior distributions using SLR projections compiled by Edwards et al. (2021) for a number of ice sheets and glaciers models, over various RCP and SSP scenarios. Using the models' outputs, we apply equation 9 to estimate the $\Lambda_{\mathrm{gis1}}$ and $\Lambda_{\mathrm{gis3}}$ parameters, equation 10 for the $\Gamma_{\mathrm{gla1}}$, $\Gamma_{\mathrm{gla3}}$ and $\gamma_{\mathrm{gla}}$ parameters, and equation 12 for the $\Lambda_{\mathrm{ais,smb}}$ and $\alpha_{\mathrm{ais}}$ parameters. During these fits, all other parameters are assumed to take their default value if structural, and their best-guess value otherwise. Results of this calibration on the individual models 315 compiled by Edwards et al. (2021) are shown for each SLR contribution in Figures A6, A7 and A8.

### 3.2.3 Ocean carbon

The ocean carbon cycle module has a number of structural parameters: $\alpha_{\mathrm{dic}}$, all $\alpha_{o,j}$ and all $\tau_{o,j}$ are taken from Strassmann and Joos (2018, Tables A2 and A3, based on the Princeton model). The prior adjustment factor $\kappa_{\tau_o}$ is arbitrarily taken to add a $20\%$ uncertainty on the oceanic transport timescales. All other priors for this module's parameters are derived from 12 CMIP6 320 models with interactive carbon cycle that contributed to C4MIP (Arora et al., 2020). $T_o$ is taken from the *piControl* simulation. $\nu_{\mathrm{gx}}$ and $\gamma_{\mathrm{gx}}$ are calibrated by applying equation 18 to the models' outputs for the *1pctCO2*, *1pctCO2-rad* and *1pctCO2-bgc* experiments, while $\beta_{\mathrm{dic}}$ and $\gamma_{\mathrm{dic}}$ are calibrated by applying equations 14-17 and 19. Results of this calibration on the individual CMIP6 models is shown in Figures A1 and A2.

### 3.2.4 Ocean acidification

In this version of Pathfinder, $\kappa_{\mathrm{pH}}$ is a structural parameter set to 1.

### 3.2.5 Land carbon

Parameters related to the passive soil carbon pool are taken from He et al. (2016, Table S5): $\nu_{\mathrm{rh3}}$ is structural, while $\alpha_{\mathrm{pass}}$ is not. All the priors for the parameters related to the preindustrial steady-state of the land carbon (i.e. $F_{\mathrm{npp0}}$, $\nu_{\mathrm{fire}}$, $\nu_{\mathrm{harv}}$, $\nu_{\mathrm{mort}}$, $\nu_{\mathrm{stab}}$, $\nu_{\mathrm{rh1}}$ and $\nu_{\mathrm{cs}}$) are derived from 11 TRENDYv7 models (Sitch et al., 2015; Le Quéré et al., 2018), exactly as for OSCAR 330 v3.1 (Gasser et al., 2020) except that all biomes and regions are lumped together. The priors for the remaining parameters are derived from 12 CMIP6 models that contributed to C4MIP (Arora et al., 2020). Using the models' outputs for the *1pctCO2*, *1pctCO2-rad* and *1pctCO2-bgc* experiments, we calibrated $\beta_{\mathrm{npp}}$, $\alpha_{\mathrm{npp}}$ and $\gamma_{\mathrm{npp}}$ through equation 24, $\beta_{\mathrm{fire}}$ and $\gamma_{\mathrm{fire}}$ through equation 28, and $\beta_{\mathrm{rh}}$ and $\gamma_{\mathrm{rh}}$ through equation 37. Results of this calibration on the individual CMIP6 models is shown in Figure A3, A4 and A5.





### 3.2.6 Permafrost carbon

The permafrost module's prior and structural parameters are recalibrated using the same algorithm as used by Gasser et al. (2018), but adapted to the global formulation of Pathfinder. First, the algorithm is run once to obtain a set of parameters reproducing the behavior of the multi-model global average of five permafrost models (with data from UVic (MacDougall, 2021) added to the four original models). This gives the values of the structural parameters (i.e. $\alpha_{\text{lst}}$, $\gamma_{\text{rt1}}$, $\gamma_{\text{rt2}}$, $\kappa_{\text{rt}}$, $a_{\min}$, all $\alpha_{\text{th},j}$, all $\tau_{\text{th},j}$, $\nu_{\text{thaw}}$ and $\nu_{\text{froz}}$). Second, the algorithm is run five additional times, for each of the five permafrost models separately, with the structural parameters established in the first step, to obtain prior distributions for the remaining parameters (i.e. $C_{\text{fr0}}$, $\kappa_a$, $\gamma_a$ and $\kappa_{\tau_{\text{th}}}$).

### 3.2.7 Atmospheric CO2

The conversion factor $\alpha_C$ is a structural parameter whose value is taken from the latest GCBs (e.g. Le Quéré et al., 2018). The prior distribution of preindustrial CO2 concentration ($C_{\text{pi}}$) is taken from the AR6 (Gulev et al., 2021, Section 2.2.3.2.1), assuming the difference between minimum and maximum over the 0–1850 period is representative of the 90% uncertainty range.

### 3.2.8 Historical CO2 and GMST

The structural $X_\mu$ and $X_\sigma$ time series are taken from the latest observations, as follows. $T_\mu$ and $T_\sigma$ are taken as the average and standard deviation of 5 observational GMST data sets: HadCRUT5 (Morice et al., 2021), Berkeley Earth (Rohde et al., 2013; Rohde, 2013), GISTEMP (Hansen et al., 2010), NOAAGlobalTemp (Huang et al., 2020), and JMA. We use the 1850–1900 period to define our preindustrial baseline, and GMST is assumed to be zero before the earliest date available in each data set. Regarding atmospheric CO2, $C_\mu$ is taken as the global value reported by NOAA/ESRL (Tans and Keeling, 2010) and $C_\sigma$ as a constant $\pm 1$ ppm uncertainty, for 1980 onward (this uncertainty is arbitrarily taken higher than the actual uncertainty estimated through instrumental measures to have more degrees of freedom in the calibration). Before that, $C_\mu$ comes from the IPCC AR6 (Dentener et al., 2021, Table AIII.1a), and $C_\sigma$ is linearly interpolated backwards from the instrumental uncertainty in 1980 to the preindustrial one (Gulev et al., 2021) in 1750. Finally, the prior distribution of $\rho_X$ is set to Uniform over $[0, 1]$, that of $\tilde{\sigma}_X$ is a unit Normal distribution, and that of $\epsilon_X$ is set arbitrarily to a Half-Normal of parameter 0.05 K for GMST and 0.5 ppm for CO2.

## 3.3 Constraints

We use a set of 19 constraints related to all aspects of the model. Many of the constraints are observations, but some are ranges assessed by expert panels such as the Global Carbon Project or the IPCC. They cover either a recent point in time or an assumed preindustrial equilibrium, and they are typically taken over a period of at least a few years to reduce the effect of natural variability. During the Bayesian assimilation, the Pathfinder model is run solely over the historical period (from 1750 to 2021). Table 1 summarises these constraints, the periods over which they are considered, and their distributions.





### 3.3.1 Climate system

To constrain the temperature response, we use the same five data sets of observed GMST as in Section 3.2.8, to derive average and standard deviation of two constraints: the average GMST change, and the average GMST yearly trend obtained through second-order accuracy gradient (Fornberg, 1988), both over the latest 20 years of data (2002–2021). Because this data is already
used as input to the Bayesian setup, albeit in a different way, it does not provide much of a constraint, and is used mostly to ensure the $\tilde{\sigma}_T$ and $\epsilon_T$ parameters remain within sensible range.

To further constrain the climate system, we use the mean OHU assessed by the IPCC AR6 over 2006–2018 (Gulev et al., 2021, Table 2.7), and the non-CO2 ERF (averaged over 2010-2019) also estimated for the AR6. The central value of the latter is taken from Dentener et al. (2021, Table AIII.3, and corresponding GitHub repository), and its uncertainty is constructed
using data from Forster et al. (2021, Table 7.8) and assuming the ERF of all species are normally distributed and uncorrelated, but fully correlated in time for each separate species (which likely overestimates the uncertainty).

To better align with the IPCC AR6, we also constrain the ECS of our model (i.e. the $T_{2\times}$ parameters). To do so, because the distribution of ECS cannot be assumed normal, we follow the framework of Roe and Baker (2007) who define the climate feedback factor ff so that $T_{2\times} = T_{2\times}^*/(1-\text{ff})$, where $T_{2\times}^*$ is the minimal ECS value (roughly corresponding to the Planck
feedback). We assume this feedback factor follows a logit-normal distribution, which implies $\text{logit}(\text{ff}) = \ln(\text{ff}/(1-\text{ff})) = \ln(T_{2\times}/T_{2\times}^* - 1)$ follows a normal distribution. We therefore constrain $\text{logit}(\text{ff})$, using distribution parameters and a value of $T_{2\times}^*$ calibrated to fit the probabilistic ranges of ECS provided by the AR6. This fit of the ECS distribution is illustrated in Figure A9.

### 3.3.2 Carbon cycle

Similarly to what is done with GMST, we constrain the atmospheric CO2 level over the latest 10 years of data (2012-2021) using the NOAA/ESRL data (Tans and Keeling, 2010). The rest of the global CO2 budget is constrained using the 2021 Global Carbon Budget (GCB; Friedlingstein et al., 2022). We use namely the net atmospheric CO2 growth and total anthropogenic emissions (fossil and land use) over the 10 years, and the ocean and land carbon sinks accumulated since the beginning of the instrumental measurement period (1960-2020). Note that our definition of the land carbon sink ignoring land use change is
consistent with that of the GCB.

Given its number of parameters and their inconsistent sources, we further constrain the land carbon module by considering present-day (mean over 1998-2002) NPP (Ciais et al., 2013; Zhao et al., 2005), and preindustrial vegetation and soil carbon pools. These preindustrial pools are taken from the AR6 for the central value (Canadell et al., 2021, Figure 5.12), but their relative uncertainty is taken from the AR5 (Ciais et al., 2013, Figure 6.1) since it is lacking in the AR6. In addition, the soil
carbon pool constraint is corrected downward by estimates of peatland carbon (Yu et al., 2010, Table 1), since it is an ecosystem missing in TRENDY models (and in ours) but not in the IPCC assessments.





### 3.3.3 Sea level rise

To constrain the separate SLR contributions from thermal expansion, GIS, AIS and glaciers, we use the model-based SLR speed estimates over the recent past (averaged over 2006–2018) reported in the AR6 (Fox-Kemper et al., 2021, Table 9.5). To

400 constrain the total contribution, we also use the historical (1901–1990) sea level rise inferred from tide gauges from the same source, although the value is corrected for the missed impact of uncharted glaciers (Parkes and Marzeion, 2018).

Contrarily to all other modules, the SLR module is not assumed to start at steady-state in 1750, which is represented through the $\lambda_{\mathrm{ice}}$ (ice $\in$ [gla, gis, ais]) parameters. We assume this is entirely due to the so-called little ice age (LIA) relaxation, which we assume can be simply modeled in Pathfinder through exponential decay of our three ice-related contributions since $t_0 = 1750$.

This gives a net LIA contribution of $H_{\mathrm{lia}} = \sum_{\mathrm{ice}} \lambda_{\mathrm{ice}} \, \tau_{\mathrm{ice}} \, \exp\!\left(-\frac{t-t_0}{\tau_{\mathrm{ice}}}\right)$. We constrain this diagnostic variable using the global SLR reported by Slangen et al. (2016) over 1900–2005 for their control experiment.

## 4 Results and diagnostics

### 4.1 Posterior distributions

Figure 5 shows the prior and posterior distributions of the model's parameters, while Figure 6 shows those of the constraints.

Figure 7 also displays the correlation matrix of the posterior parameters. (There is no correlation among the prior parameters.)

### 4.1.1 Climate system

Our climate-related constraints lead to adjusting all the parameters of the climate module. As explained in Section 3.2.8, the constraints for present-day GMST change and its derivative are met by construction.

The ECS ($T_{2\times}$) is the parameter with the strongest adjustment, since it is directly constrained. Its precise value is discussed

hereafter in Section 4.3, but we note that it is unsurprisingly decreased, as the CMIP6 model ensemble tends to overestimate the ECS compared to the IPCC assessed value. Consequently, our posterior $\mathrm{logit}(\mathrm{ff})$ matches well the constraint. The adjustment of the ECS significantly reduces the gap between our posterior distribution of the non-CO2 ERF and its constraint, although the posterior central value remains 41% lower (but well within uncertainty range).

Among the dynamic parameters that are adjusted, we note that the deep ocean heat capacity $\Theta_d$ is somewhat increased

compared to the prior, and the heat exchange coefficient $\theta$ is also increased. These dynamic parameters are likely adjusted through our OHU constraint that is corrected in the posterior so the difference in the central values is lowered from 22% to 14%, which remains well within the uncertainty range.

In addition, a number of weak but physically meaningful correlations across the climate module's parameters are found, such as a positive correlation between $T_{2\times}$ and $\epsilon_{\mathrm{heat}}$ (see e.g. Geoffroy et al., 2013a), a positive correlation between $T_{2\times}$ and

425 $\Theta_d$ (that tends to exclude configuration that would warm fast and high), and a negative correlation between $T_{2\times}$ and $\phi$ (to match the GMST and ERF constraints together).



### 4.1.2 Carbon cycle

Similarly to GMST, the posterior distribution of atmospheric CO2 concentration matches the constraint by construction. It's
derivative, however, is (slightly overly) corrected to match the GCB estimate. Global anthropogenic CO2 emissions are signif-
430 icantly increased to get closer to the GCB constraint, but remain 9% too low. Since these emissions are determined through
mass balance and the atmospheric CO2 matches observations, this implies that the total carbon sinks (i.e. $F_{\text{land}} + F_{\text{ocean}}$) must
be weaker.

This is confirmed for the ocean sink, as the posterior $F_{\text{ocean}}$ is 8% lower than the constraint, but still noticeably corrected if
compared to the prior. This correction is explained by small adjustments in some parameters of the ocean carbon module. The
435 mixed layer depth is slightly increased through $\beta_{\text{dic}}$. All other parameters remain mostly unaffected by the calibration, and only
minor correlations are found. These results, along with the fact that our prior distribution spans only about half of the constraint
distribution, suggest that there is a structural limitation in our ocean carbon module that warrants further investigation.

It is also confirmed that the posterior land sink is weaker than the constraint, by 15%, which is nevertheless a significant
reduction of the prior gap of 34%. To explain this adjustment, we observe that the CO2-fertilization sensitivities ($\beta_{\text{npp}}$ and
440 $\gamma_{\text{npp}}$) are adjusted upwards. However our constraint on present-day NPP prevents these adjustments to be too important, as
the posterior distribution of this variable is similar to the prior and remains 8–9% higher than its constraint. An increased
preindustrial NPP mechanically leads to an increase in preindustrial carbon pools, but these require further adjustments of the
land carbon turnover rates, and most notably the mortality rate $\nu_{\text{mort}}$ and the passive carbon fraction $\alpha_{\text{pass}}$, to better match
their constraints (of which the one on total soil carbon is perfectly met).

The land carbon module exhibits significant correlations among posterior parameters. This is likely a consequence of all
the constraints combined as they dictate both the preindustrial steady-state of the module and it's transient response over the
historical period. Eliminated configurations are those, for instance, that would show high initial carbon pools that are very
sensitive to climate change (as these would lead to a very weak land sink), or that would exhibit a weak CO2-fertilization effect
associated with a fast turnover time (that would also lead to a weak sink).

### 450 4.1.3 Sea level rise

The prior parameters of the SLR module are the least informed of this Bayesian setup. The model initially underestimates
the thermal expansion, as well as the GIS and AIS SLR rates. The calibration brings the posterior distributions closer to their
respective constraints but it always remain in the lower end of the uncertainty range. The correction is done by adjusting many
of the module's parameters (most notably $\Lambda_{\text{gis1}}$, $\Lambda_{\text{ais,smb}}$, $\lambda_{\text{ais}}$ or $\lambda_{\text{gla}}$), and by finding strong correlations among them (thus
excluding physically unrealistic combinations).

The historical SLR is markedly corrected by the constraint: from a 19% gap between the constraint and the prior estimate,
to only 7% after calibration. Here, we also note that the some of individual contributions to historical SLR reported in AR6 do
not match that total SLR (Fox-Kemper et al., 2021, Table 9.5), which likely has some impact on the consistency between our
constraints. Finally, although the LIA relaxation contribution is not altered by the calibration, and remains 50% too high, it is





the likely source of the strong correlations found among the parameters of this module, because it straightforwardly links the individual SLR contributions together.

## 4.2    Historical period

Because in the Bayesian setup we do not use annual time series of observations as constraints, the posterior distributions given in Figure 6 do not inform on the whole dynamic of the model over the historical period. To further diagnose the model's

behavior, Figure 8 gives the time series from 1900 to 2021 of six key variables. GMST and atmospheric CO2 match very well the historical observations, by construction of these input time series. The non-$CO_2$ ERF exhibits a very high variability, owing to our temperature-driven setup and the natural variability in the GMST input. Beyond that, the ERF time series is consistent with the AR6 estimates (Smith et al., 2021), albeit somewhat lower on average in the recent past, as seen with the posterior distribution. Consistently with the interpretation of carbon cycle variables in the calibration results, anthropogenic

CO2 emissions, and the ocean and land carbon sinks are slightly underestimated compared to the GCB estimates (Friedlingstein et al., 2022). Several reasons could explain this discrepancy, from the lack of land use change in Pathfinder to the inconsistency of the GCB figures (that do not close the budget, while ours do). Nevertheless, the interest of the calibration is clearly illustrated, as the posterior uncertainty range covers observations much better than the prior one.

## 4.3    Idealized simulations

To complete the diagnosis of our model with common metrics used with climate and carbon models, we ran a set of standard idealized experiments, corresponding to the CMIP6 *abrupt-2xCO2*, *1pctCO2*, *1pctCO2-bgc* and *1pctCO2-rad*. A summary of these metrics' values is given in Table 2, and the resulting time series are shown in Figure 9.

The *abrupt-2xCO2* experiment sees an abrupt doubling of atmospheric CO2, and it is used to diagnose the model's ECS that is defined as the equilibrium temperature for a doubling of the preindustrial atmospheric concentration of CO2 (we acknowl-

edge that it is superfluous with this version of Pathfinder since it is also a parameter). Using the GMST anomaly at the end of 1500 years of this experiment leads to an unconstrained estimate of ECS of 4.1 ± 1.3 K and a constrained estimate of 3.3 ± 0.7 K. Consistently, the latter value is between the ECS value extracted from CMIP6 models (Meehl et al., 2020) that is higher (3.7 ± 1.1 K) and the final value assessed in the AR6 that is lower (3.0 K, with a 67% confidence interval between 2.5 and 4.0 K).

Using the *1pctCO2* experiment that sees a 1% yearly increase in atmospheric CO2, we can estimate the model's transient climate response (TCR) that is defined as the GMST change after 70 years, when atmospheric concentration CO2 has just doubled. The CMIP6 models have a TCR of 2.0 ± 0.4 K (Meehl et al., 2020). Pathfinder's unconstrained value is higher, at 2.2 ± 0.5 K, while the constrained one goes down to 1.9 ± 0.3 K. If we divide the TCR by the cumulative anthropogenic CO2 emissions compatible with the atmospheric CO2 increase in this experiment, we obtain an estimate of the transient climate

response to emissions (TCRE). Similarly to the TCR, it is higher in the unconstrained ensemble and lower in the constrained one, when compared to CMIP6 models (Arora et al., 2020). Both downward adjustments of the TCR and TCRE are consistent with that of ECS, with the posterior TCRE matching very well the AR6 assessed range (Canadell et al., 2021).





To look more closely at the carbon cycle, we perform two variants of the latter experiment: in *1pctCO2-rad*, atmospheric CO2 only has a radiative effect, as it is kept at preindustrial level for the carbon cycle; whereas in *1pctCO2-bgc*, atmospheric

CO2 only has a biogeochemical effect, as the climate system sees only preindustrial level. These three experiments are used to calculate the carbon-concentration ($\beta$) and carbon-climate ($\gamma$) feedback metrics that measure the carbon sinks' sensitivities to changes in atmospheric CO2 and GMST, respectively. We apply the same method as Arora et al. (2020) to calculate these, which leads to metrics at the time of CO2 doubling that are in line with CMIP6 models (Arora et al., 2020). As both carbon sinks were adjusted upwards by the Bayesian calibration, the constraints logically increased both $\beta_{\mathrm{ocean}}$ and $\beta_{\mathrm{land}}$, to values

fairly close to those of the complex models. The $\gamma_{\mathrm{ocean}}$ is not affected by the calibration, and remains 45% too low, which again suggests a structural limitation in our formulation of the ocean sink. This is however compensated during calibration by the $\gamma_{\mathrm{land}}$ being 26% higher than in complex models.

### 4.4    Scenarios

To further validate Pathfinder, we run the five SSP scenarios (Riahi et al., 2017) for which climate and carbon cycle projections

were reported by a large-enough number of models in the AR6 (namely, *ssp119*, *ssp126*, *ssp245*, *ssp370* and *ssp585*). These simulations are run with prescribed CO2 concentration and non-CO2 ERF (the latter is taken from Smith et al. (2021)). Time series of GMST and cumulative land and ocean sinks are shown on Figure 9. Table 3 shows a comparison of the projected changes in GMST to the CMIP6 estimates (Lee et al., 2021, Table 4.2), and of carbon pools to Liddicoat et al. (2021) (since this was not directly reported in the AR6).

Be it on short-, mid- or long-term, Pathfinder's projections of GMST are very much in line with the one assessed by the IPCC in the AR6 based on multiple lines of evidence (Lee et al., 2021, Table 4.5). The only significant difference is a smaller uncertainty range in our projections for the longer-term periods. Although this is the result of the efficiency of the Bayesian calibration, one might wonder whether the climate module is over-constrained (or equivalently, too limited in its number of parameters).

The ocean carbon storage appears overestimated by 5% to 20% by Pathfinder across SSP scenarios. This is consistent with the upward adjustment of the ocean carbon sink stemming from our Bayesian calibration. To compare the land carbon storage with CMIP6 models, because our land carbon module does not include land use change processes, we correct the value reported by complex models by the cumulative land use change emissions of each scenario (Riahi et al., 2017; Gidden et al., 2019). While the land carbon storage of Pathfinder is well in line under *ssp126* (a scenario consistent with the 2 °C target), it is

underestimated in *ssp119* (consistent with the 1.5 °C target), and increasingly overestimated in higher warming scenarios. A likely explanation is that the climate-carbon feedback on land is underestimated in Pathfinder, as suggested by the $\gamma$ metric seen in Section 4.3. Alternatively (or concurrently), the absence of loss of sink capacity caused by land cover change (Gasser and Ciais, 2013; Gasser et al., 2020) can explain the overestimation of the land sink under high CO2. The Pathfinder model's estimates of both sinks remain nonetheless well within the CMIP6 models' uncertainty ranges.

Our SLR emulator gives estimates (Table 4) that are always on the lower of the range reported in the AR6 (Fox-Kemper et al., 2021, Table 9.9). This can be explained by the fact that our individual SLR rate estimates are on the lower end of





their respective constraints (see Section 4.1.3). This discrepancy also highlights potential structural limitations in the SLR module (e.g. too few separate contributions), and the difficulty of calibrating the module given the short time period of data available, both from complex models (that cover the 21st century only) and observations, compared to the long time scale of

the SLR processes. Nevertheless, our estimates remain within uncertainty range of the IPCC assessment, especially as we do not account for contribution from land water storage that causes an additional 0.03 [0.01, 0.04] m of SLR in all scenarios in 2100 (Fox-Kemper et al., 2021).

## 5  Concluding remarks

In this paper, we have presented the Pathfinder model: a simple global carbon-climate model with selected impact variables,
carefully designed to balance accuracy of representation and simplicity of formulation, and calibrated through Bayesian inference on the latest data from Earth system models and observations. Pathfinder has been shown to perform very well in comparison to complex models, although there remains room for further improvement of the model and its calibration setup. We identify four main avenues to improve the model.

First, the ocean carbon module appears to be limited by its structure inherited from a 25-year-old (yet seminal) article (Joos
et al., 1996). Although it is undoubtedly a significant undertaking, developing alternative formulations of the ocean carbon dynamic, calibrated on state-of-the-art ocean models and properly interacting with the ocean of the climate module, would benefit more than just the SCM community.

Second, integration of land use and land cover change in such a model appears warranted, despite the difficulty of doing so in a physically sensible yet simple manner. Given our expertise with the OSCAR model and its bookkeeping module (Gasser
et al., 2020), we are confident that this can be done, although it will demand extra care to keep the model compatible with the IAMs it is meant to be linked to.

Third, the Bayesian setup can be extended, notably by including more time periods for the existing constraints, but also by introducing and constraining entirely new variables such as isotopic ratios (Hellevang and Aagaard, 2015) or inter-hemispheric gradients (Ciais et al., 2019); although a balance must be stricken with respect to the calibration's computation time. Here, we
caution against including complex models' results as constraints in the Bayesian calibration, as was done for the IPCC AR6 (Smith et al., 2021; Nicholls et al., 2021), as it goes against the philosophy of Pathfinder that is to use complex models as prior information and only real world observations (or assessment combining many lines of evidence) as constraints.

Fourth, although our model is restricted to CO2 by design because of how IAMs like DICE (Nordhaus, 2017) are also limited to CO2 emissions, we can imagine many reasons why one would want to add non-CO2 climate forcers into Pathfinder. We
would suggest doing so by following the model's philosophy: that is, by taking existing reduced-complexity formulations such as something between FaIR (Leach et al., 2020) and OSCAR (Gasser et al., 2017), and adding the new parameters into the Bayesian setup with the relevant observational constraints.

In spite of these few shortcomings and potential development leads, Pathfinder v1.0 is a powerful tool that fits perfectly the niche it has been created for. We will further demonstrate the strengths and flexibility of Pathfinder in other publications.





Meanwhile, we invite the community to seize this open source model, and use it in any study that could benefit from a simple, fast and accurate carbon-climate model, aligned with the latest climate science.



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





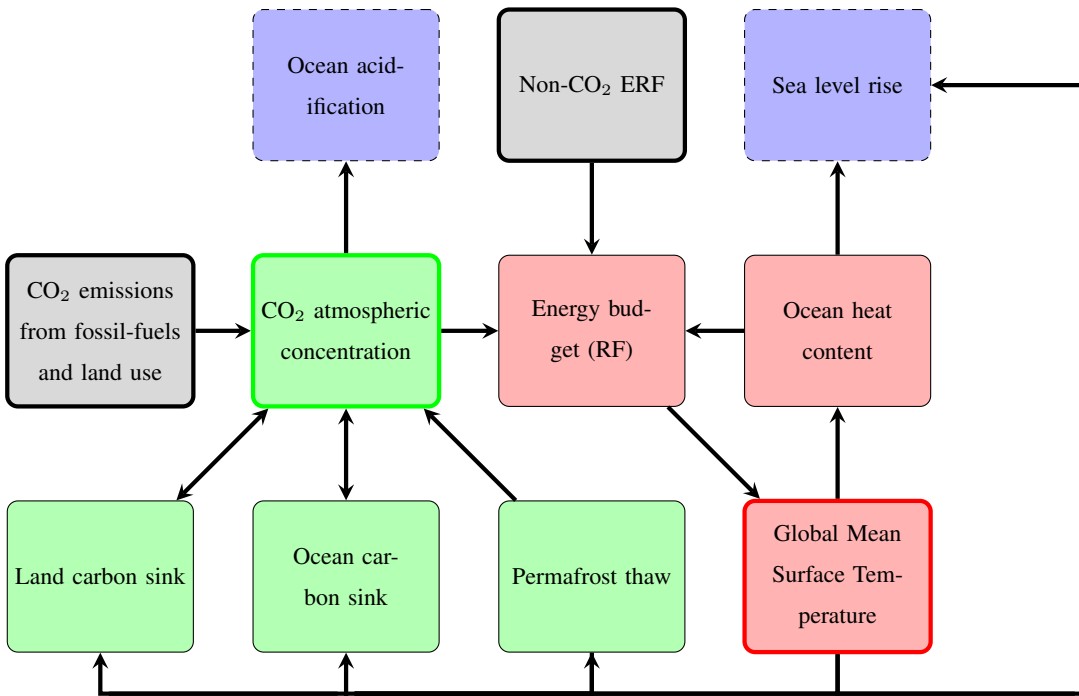

**Figure 1.** Pathfinder in a nutshell: Green blocks represent the carbon cycle, and red blocks the climate response. Blue blocks with dotted arrows are impacts that can be derived with the model. Grey blocks are variables that are directly related to anthropogenic activity. Possible inputs of the model are distinguishable through the bold contours of the blocks. In this scheme, arrows correspond to a forward mode where inputs would be $E_{\mathrm{CO2}}$ and $R_x$





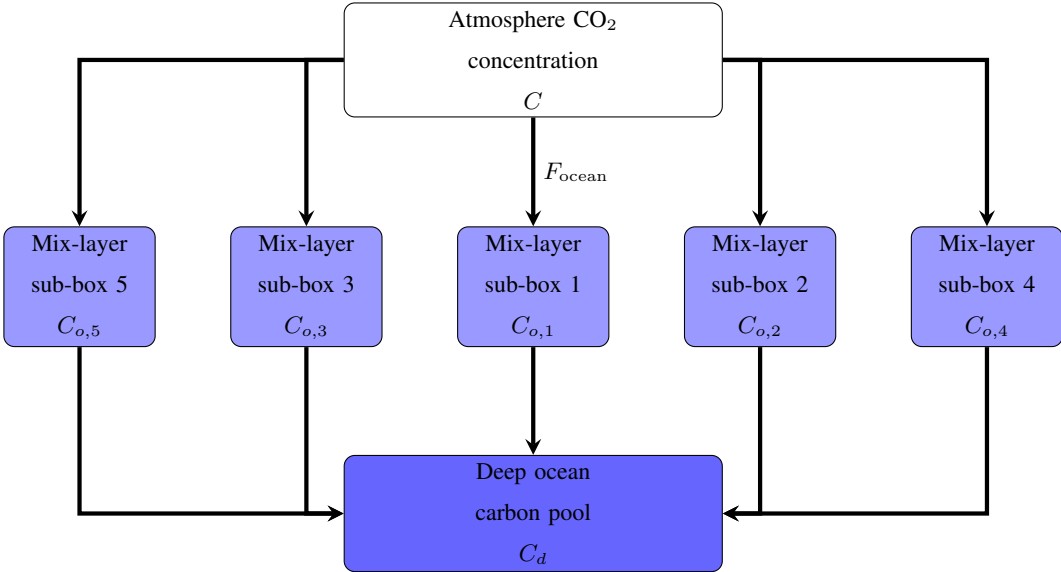

**Figure 2.** The ocean sink model in Pathfinder follows the structure of the mixed-layer pulse response function introduced by Joos et al. (1996). The mix-layer is represented through five subpools which each has a different timescale for transport to the deep ocean carbon pool





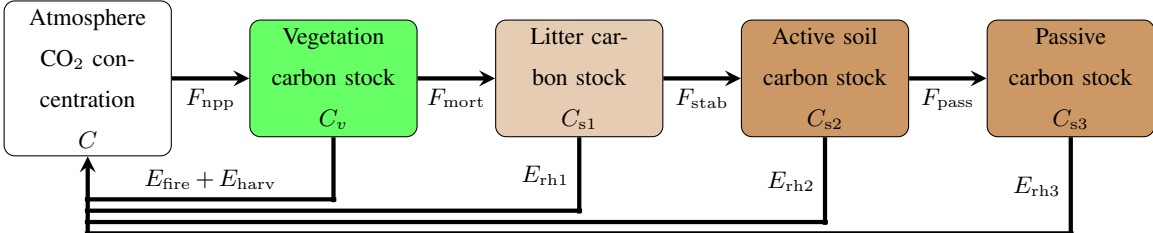

**Figure 3.** The land sink model in Pathfinder is derived from OSCAR (Gasser et al., 2017) and represents the biosphere as a set of four carbon pools: vegetation, litter , active soil and passive soil. These pools exchange carbon through fluxes whose direction is given by the arrows.





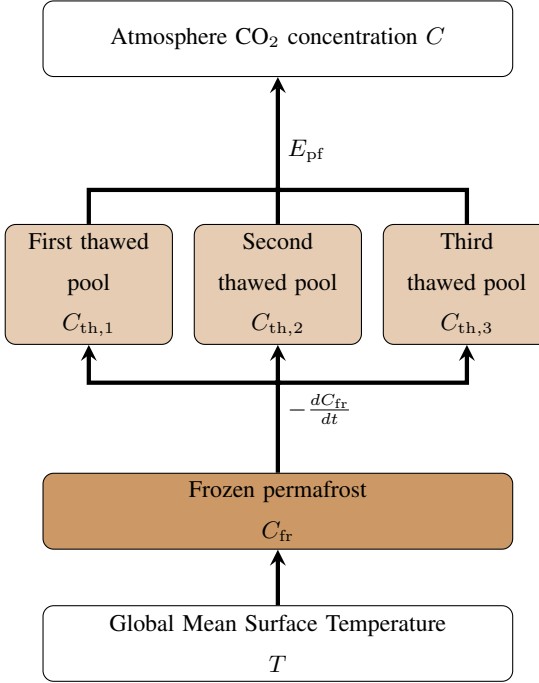

**Figure 4.** The permafrost carbon model in Pathfinder is taken from Gasser et al. (2018). The frozen pool dynamic lags behind a theoretical value that is determined by the temperature anomaly. Thawed carbon is then split between three pools that are emitted to the atmosphere at different rates.



| Variable | Period | Method | Prior | Posterior | Constraints | Unit |
|---|---|---|---|---|---|---|
| $E_{CO2}$ | 2011-2020 | Mean | 9.1 ± 1.3 | 10.0 ± 0.7 | 11.0 ± 0.9 | PgC yr$^{-1}$ |
| $\frac{dC}{ct}$ | 2011-2020 | Mean | 2.41 ± 0.06 | 2.40 ± 0.01 | 2.40 ± 0.01 | ppm yr$^{-1}$ |
| $F_{land}$ | 1960-2020 | Sum | 95 ± 52 | 123 ± 26 | 145 ± 35 | PgC |
| $F_{ocean}$ | 1960-2020 | Sum | 89± 12 | 97 ± 13 | 105 ± 20 | PgC |
| $C_v$ | 1750 | Mean | 407 ± 54 | 407 ± 37 | 450 ± 50 | PgC |
| $C_s$ | 1750 | Mean | 1181 ± 735 | 1086 ± 284 | 1088 ± 249 | PgC |
| $F_{NPP}$ | 1998-2002 | Mean | 60.0 ± 7.9 | 59.5 ± 3.9 | 55.0 ± 5.0 | PgC yr$^{-1}$ |
| $C$ | 2012-2021 | Mean | 403.6 ± 0.3 | 403.6 ± 0.1 | 401.2 ± 0.1 | ppm |
| $R_x$ | 2010-2019 | Mean | 0.01 ± 0.47 | 0.33 ± 0.37 | 0.56± 0.53 | W m$^{-2}$ |
| $T$ | 2001-2020 | Mean | 0.96 ± 0.08 | 0.97 ± 0.06 | 1.00 ± 0.07 | K |
| $\frac{dT}{dt}$ | 2000-2019 | Mean | 0.028 ± 0.003 | 0.028 ± 0.002 | 0.029 ± 0.002 | K yr$^{-1}$ |
| $\frac{dU_{ohc}}{dt}$ | 2006-2018 | Mean | 0.56 ± 0.10 | 0.62 ± 0.09 | 0.72 ± 0.17 | W m$^{-2}$ |
| $\frac{dH_{thx}}{dt}$ | 2006-2018 | Mean | 1.02 ± 0.22 | 1.14 ± 0.21 | 1.39 ± 0.40 | mm yr$^{-1}$ |
| $\frac{dH_{gla}}{dt}$ | 2006-2018 | Mean | 0.63 ± 0.24 | 0.62 ± 0.04 | 0.62 ± 0.03 | mm yr$^{-1}$ |
| $\frac{dH_{ais}}{dt}$ | 2006-2018 | Mean | -0.02 ± 0.23 | 0.30 ± 0.10 | 0.37 ± 0.08 | mm yr$^{-1}$ |
| $\frac{dH_{gis}}{dt}$ | 2006-2018 | Mean | 0.36 ± 0.12 | 0.57 ± 0.10 | 0.63 ± 0.07 | mm yr$^{-1}$ |
| $H_{tot}$ | 1901-1990 | Difference | 72 ± 17 | 83 ± 10 | 89 ± 32 | mm |
| $H_{lia}$ | 1750 | Mean | 45 ± 17 | 45 ± 11 | 30 ± 13 | mm |
| $logit(ff)$ | 1750 | Mean | 1.69 ± 0.38 | 1.47 ± 0.28 | 1.38 ± 0.37 | 1 |

**Table 1.** Constrained variables in Pathfinder, with values before and after calibration. Variables are noted under their code notation, and Tables A1 and A2 provide the corresponding notation in text. The uncertainty correspond to the 1 $\sigma$ uncertainty range.

| $2\times CO_2$ | Pathfinder unconstrained | Pathfinder constrained | CMIP6 | AR6 |
|---|---|---|---|---|
| ECS (K) | 4.1 ± 1.3 | 3.3 ± 0.7 | 3.7 ± 1.1 | 3.0 (2.0, 4.5) |
| TCR (K) | 2.2 ± 0.5 | 1.9 ± 0.3 | 2.0 ± 0.4 | 1.8 (1.4, 2.2) |
| TCRE (K EgC$^{-1}$) | 2.20 ± 0.63 | 1.65 ± 0.32 | 1.77 ± 0.37 | 1.65 (1.0, 2.3) |
| $\beta_{ocean}$ (PgC ppm$^{-1}$) | 0.81 ± 0.10 | 0.87 ± 0.11 | 0.91 ± 0.09 | |
| $\gamma_{ocean}$ (PgC K$^{-1}$) | -12.9 ± 5.4 | -12.5 ± 6.0 | -8.6 ± 2.9 | |
| $\beta_{land}$ (PgC ppm$^{-1}$) | 1.05 ± 0.5 | 1.26 ± 0.30 | 1.22 ± 0.40 | |
| $\gamma_{land}$ (PgC K$^{-1}$) | -33.2 ± 26.6 | -25.3 ± 24.2 | -34.1 ± 38.4 | |

**Table 2.** Diagnostics of climate and carbon-cycle responses in Pathfinder before and after Bayesian calibration. Comparison with AR6 (Forster et al., 2021; Canadell et al., 2021) and CMIP6 data (Arora et al., 2020; Meehl et al., 2020) is shown. For AR6 data we give the median and the 90% confidence interval while for every other values we give the mean $\pm$ 1 $\sigma$





**Figure 5.** Parameters distributions before (black lines) and after (blue lines) the Bayesian calibration. Parameters are noted under their code notation, and Tables A3 and A4 provide the corresponding notation in text.





**Figure 6.** Distributions of the constrained variables. Dashed lines give the distributions used to constrain. Black lines give the distribution before calibration while blue lines give the distribution posterior to calibration. Under a variable's name, we give the period over which the constraint is estimated, and the data processing method: mean over the period, difference between last and first year, or sum of all the years over the period. (1750 is the preindustrial.) Variables are noted under their code notation, and Tables A1 and A2 provide the corresponding notation in text.





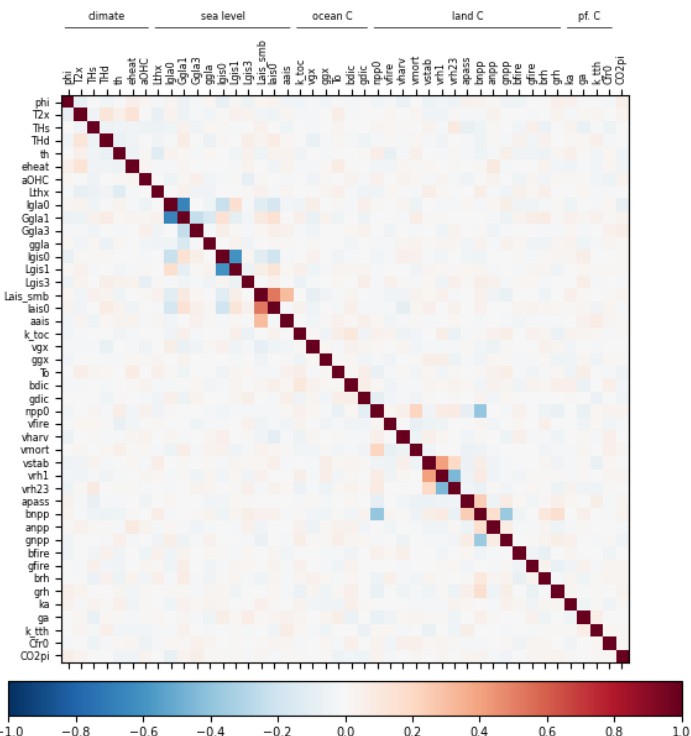

**Figure 7.** Correlation matrix of Pathfinder's parameters after the Bayesian calibration. Parameters are classified according to the equations they are related to: climate system, sea level, ocean carbon, land carbon and permafrost carbon. Parameters are noted under their code notation, and Tables A3 and A4 provide the corresponding notation in text.



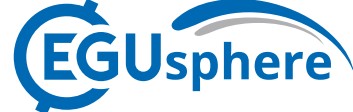

**Figure 8.** Historical time series of key variables from Pathfinder. Red lines are observations, black lines are the model's outputs before calibration, and blue lines are the same after calibration. Shaded areas and vertical bars correspond to the $1\sigma$ uncertainty range. Temperature observations are taken from HadCRUT5 (Morice et al., 2021), Cowtan and Way (2014), Berkeley Earth (Rohde et al., 2013; Rohde, 2013), GISTEMP (Hansen et al., 2010), and NOAA/MLOST (Vose et al., 2012). Other sources are NOAA/ESRL (Tans and Keeling, 2010), GCB 2021 (Friedlingstein et al., 2022), and AR6 (Smith et al., 2021).





**Figure 9.** Time series of GMST change, land carbon uptake and ocean carbon uptake for idealized experiments (*abrupt-2xCO2*, *1pctCO2*, *1pctCO2-bgc* and *1pctCO2-rad*), and projections according to SSP scenarios in PathFinder. Shaded areas give the $1\sigma$ uncertainty range.





| Experiment | Model | GMST (K) | GMST (K) | GMST (K) | Ocean Carbon Storage (PgC) | Land Carbon Storage (PgC) |
|---|---|---|---|---|---|---|
| | | 2021–2040 | 2041–2060 | 2081–2100 | 2015–2100 | 2015–2100 |
| ssp119 | Pathfinder | 1.5 (1.3, 1.8) | 1.6 (1.4, 1.9) | 1.5 (1.2, 1.7) | $132 \pm 21$ | $49 \pm 33$ |
| ssp119 | AR6 or CMIP6 | 1.5 (1.2, 1.7) | 1.6 (1.2, 2.0) | 1.4 (1.0, 1.8) | $111 \pm 11$ | $73 \pm 33$ |
| | | | | | | |
| ssp126 | Pathfinder | 1.5 (1.3, 1.8) | 1.8 (1.6, 2.1) | 1.9 (1.6, 2.2) | $179 \pm 28$ | $109 \pm 45$ |
| ssp126 | AR6 or CMIP6 | 1.5 (1.2, 1.8) | 1.7 (1.3, 2.2) | 1.8 (1.3, 2.4) | $162 \pm 8$ | $120 \pm 50$ |
| | | | | | | |
| ssp245 | Pathfinder | 1.6 (1.4, 1.8) | 2.1 (1.8, 2.4) | 2.8 (2.4, 3.3) | $265 \pm 41$ | $225 \pm 76$ |
| ssp245 | AR6 or CMIP6 | 1.5 (1.2, 1.8) | 2.0 (1.6, 2.5) | 2.7 (2.1, 3.5) | $252 \pm 11$ | $178 \pm 76$ |
| | | | | | | |
| ssp370 | Pathfinder | 1.6 (1.4, 1.8) | 2.2 (1.9, 2.6) | 3.7 (3.2, 4.3) | $354 \pm 53$ | $330 \pm 112$ |
| ssp370 | AR6 or CMIP6 | 1.5 (1.2, 1.8) | 2.1 (1.7, 2.6) | 3.6 (2.8, 4.6) | $338 \pm 15$ | $269 \pm 124$ |
| | | | | | | |
| ssp585 | Pathfinder | 1.7 (1.5, 1.9) | 2.5 (2.2, 2.9) | 4.4 (3.8, 5.2) | $420 \pm 63$ | $409 \pm 148$ |
| ssp585 | AR6 or CMIP6 | 1.6 (1.3, 1.9) | 2.4 (1.9, 3.0) | 4.4 (3.3, 5.7) | $398 \pm 17$ | $311 \pm 162$ |

**Table 3.** Comparison of SSP scenarios for GMST change projections (w.r.t. 1850–1900) to AR6 (Lee et al., 2021, Table 4.5), and for ocean and land carbon storage projections to CMIP6 (Liddicoat et al., 2021). Land carbon storage projections were corrected using the land use change emissions data from SSPs (Riahi et al., 2017; Gidden et al., 2019). For GMST data we give the median and the 90% confidence interval while for every other values we give the mean $\pm\ 1\ \sigma$



| Experiment | Model | SLR (m) 2050 | SLR (m) 2100 | SLR rate (mm yr$^{-1}$) 2040–2060 | SLR rate (mm yr$^{-1}$) 2080–2100 |
|---|---|---|---|---|---|
| ssp119 | Pathfinder | 0.15 (0.13, 0.18) | 0.30 (0.25, 0.36) | 3.5 (2.9, 4.2) | 2.7 (2.2, 3.4) |
| ssp119 | AR6 | 0.18 (0.15, 0.23) | 0.38 (0.28, 0.55) | 4.1 (2.8, 6.0) | 4.2 (2.4, 6.6) |
| ssp126 | Pathfinder | 0.16 (0.14, 0.19) | 0.35 (0.30, 0.43) | 4.1 (3.5, 5.0) | 3.6 (2.9, 4.5) |
| ssp126 | AR6 | 0.19 (0.16, 0.25) | 0.44 (0.32, 0.62) | 4.8 (3.5, 6.8) | 5.2 (3.2, 8.0) |
| ssp245 | Pathfinder | 0.17 (0.15, 0.20) | 0.46 (0.39, 0.56) | 5.0 (4.2, 6.0) | 6.2 (5.1, 8.0) |
| ssp245 | AR6 | 0.20 (0.17, 0.26) | 0.56 (0.44, 0.76) | 5.8 (4.4, 8.0) | 7.7 (5.2, 11.6) |
| ssp370 | Pathfinder | 0.18 (0.15, 0.21) | 0.56 (0.47, 0.69) | 5.5 (4.7, 6.7) | 9.1 (7.4, 11.7) |
| ssp370 | AR6 | 0.22 (0.18, 0.27) | 0.68 (0.55, 0.90) | 6.4 (5.0, 8.7) | 10.4 (7.4, 14.8) |
| ssp585 | Pathfinder | 0.19 (0.17, 0.23) | 0.67 (0.56, 0.82) | 6.4 (5.4, 7.8) | 11.4 (9.1, 15.0) |
| ssp585 | AR6 | 0.23 (0.20, 0.29) | 0.77 (0.63, 1.01) | 7.2 (5.6, 9.7) | 12.1 (8.6, 17.6) |

**Table 4.** Comparison of SSP scenarios between Pathfinder and AR6 for SLR (w.r.t. 1995–2014) and SLR speed projections (Fox-Kemper et al., 2021, Table 9.9). We give the median value and the 90% confidence interval in parentheses



**Figure A1.** Calibration to estimate prior $\nu_{\mathrm{gx}}$ and $\gamma_{\mathrm{gx}}$ from CMIP6 time series of $F_{\mathrm{ocean}}$. We fit our equation on the results of the +1% $CO_2$ (*1pctCO2*) experiment (in blue) and its variants *1pctCO2-rad* (in green) and *1pctCO2-bgc* (in orange). Coloured markers are CMIP6 models data while the solid black lines show the fit from Pathfinder. Panels without black line indicate that at least one of the required variables was not reported by the complex model.

# Appendix A





**Figure A2.** Calibration to estimate prior $\beta_{dic}$ and $\gamma_{dic}$ from CMIP6 time series of $p_{CO_2}$. We fit our equation on the results of the +1% $CO_2$ (*1pctCO2*) experiment (in blue) and its variants *1pctCO2-rad* (in green) and *1pctCO2-bgc* (in orange). Coloured markers are CMIP6 models data while the solid black lines show the fit from Pathfinder. Panels without black line indicate that at least one of the required variables was not reported by the complex model.

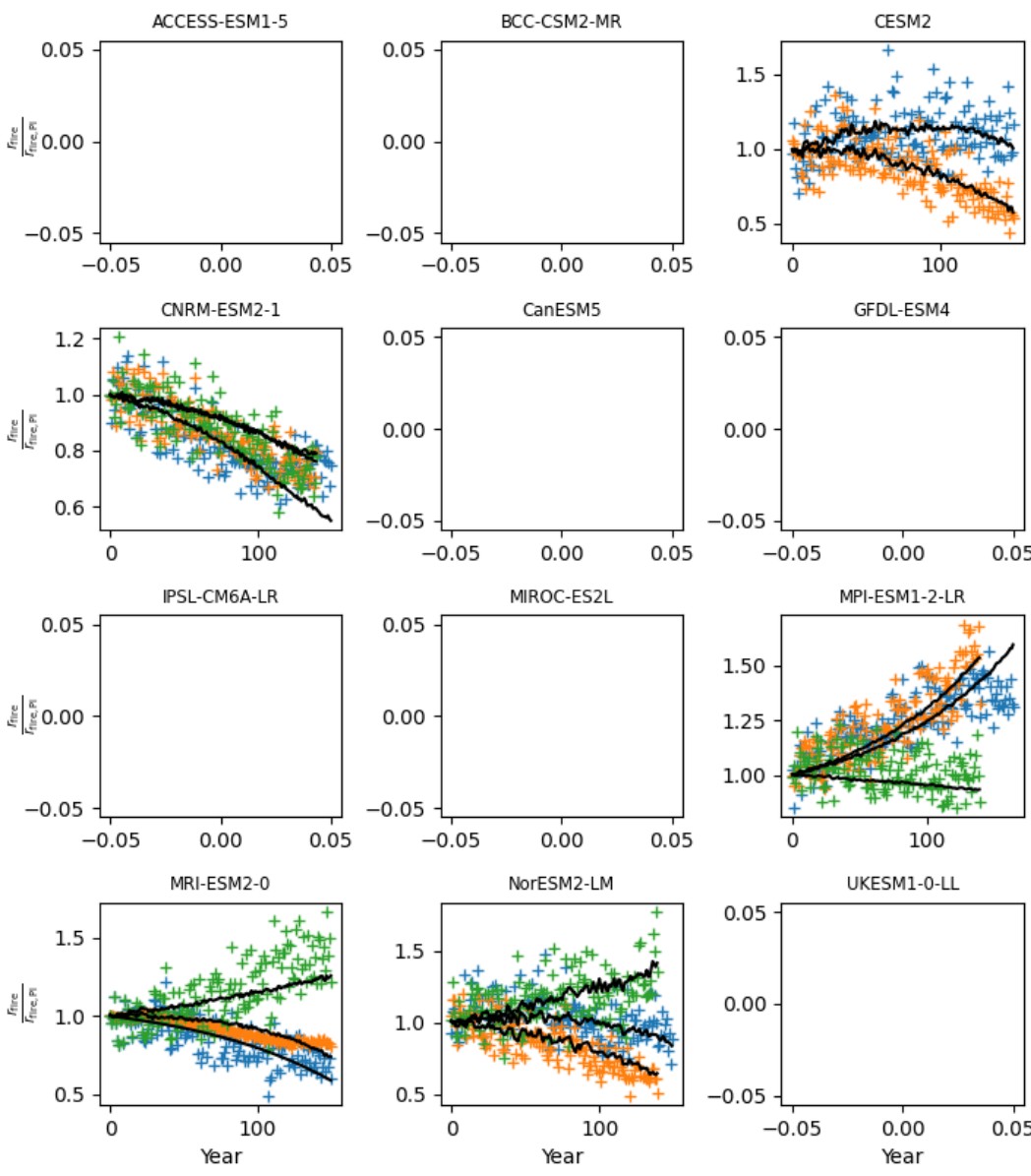

**Figure A3.** Calibration to estimate prior $\beta_{\text{ef}}$ and $\gamma_{\text{ef}}$ from CMIP6 time series of $r_{\text{fire}}$, shown as a ratio over its preindustrial value. We fit our equation on the results of the +1% $CO_2$ (*1pctCO2*) experiment (in blue) and its variants *1pctCO2-rad* (in green) and *1pctCO2-bgc* (in orange). Coloured markers are CMIP6 models data while the solid black lines show the fit from Pathfinder. Panels without black line indicate that at least one of the required variables was not reported by the complex model.







**Figure A4.** Calibration to estimate prior $\beta_{\mathrm{rh}}$ and $\gamma_{\mathrm{rh}}$ from CMIP6 time series of $r_{\mathrm{rh}}$, shown as a ratio over its preindustrial value. We fit our equation on the results of the +1% $CO_2$ (*1pctCO2*) experiment (in blue) and its variants *1pctCO2-rad* (in green) and *1pctCO2-bgc* (in orange). Coloured markers are CMIP6 models data while the solid black lines show the fit from Pathfinder. Panels without black line indicate that at least one of the required variables was not reported by the complex model.





**Figure A5.** Calibration to estimate prior $\beta_{\text{npp}}$, $\alpha_{\text{npp}}$ and $\gamma_{\text{npp}}$ from CMIP6 time series of $r_{\text{npp}}$, shown as a ratio over its preindustrial value. We fit our equation on the results of the +1% $CO_2$ (*1pctCO2*) experiment (in blue) and its variants *1pctCO2-rad* (in green) and *1pctCO2-bgc* (in orange). Coloured markers are CMIP6 models data while the solid black lines show the fit from Pathfinder. Panels without black line indicate that at least one of the required variables was not reported by the complex model.







**Figure A6.** Calibration to estimate the prior of GIS SLR module parameters ($\Lambda_{gis1}$, $\Lambda_{gis3}$ and $\lambda_{gis0}$). We fit our equation on the compiled outputs from Edwards et al. (2021) for which there is more than one scenario available. Each panel's title displays the name of the institute, model and configuration used. Coloured markers are the models data while the solid black lines show the fit from Pathfinder.





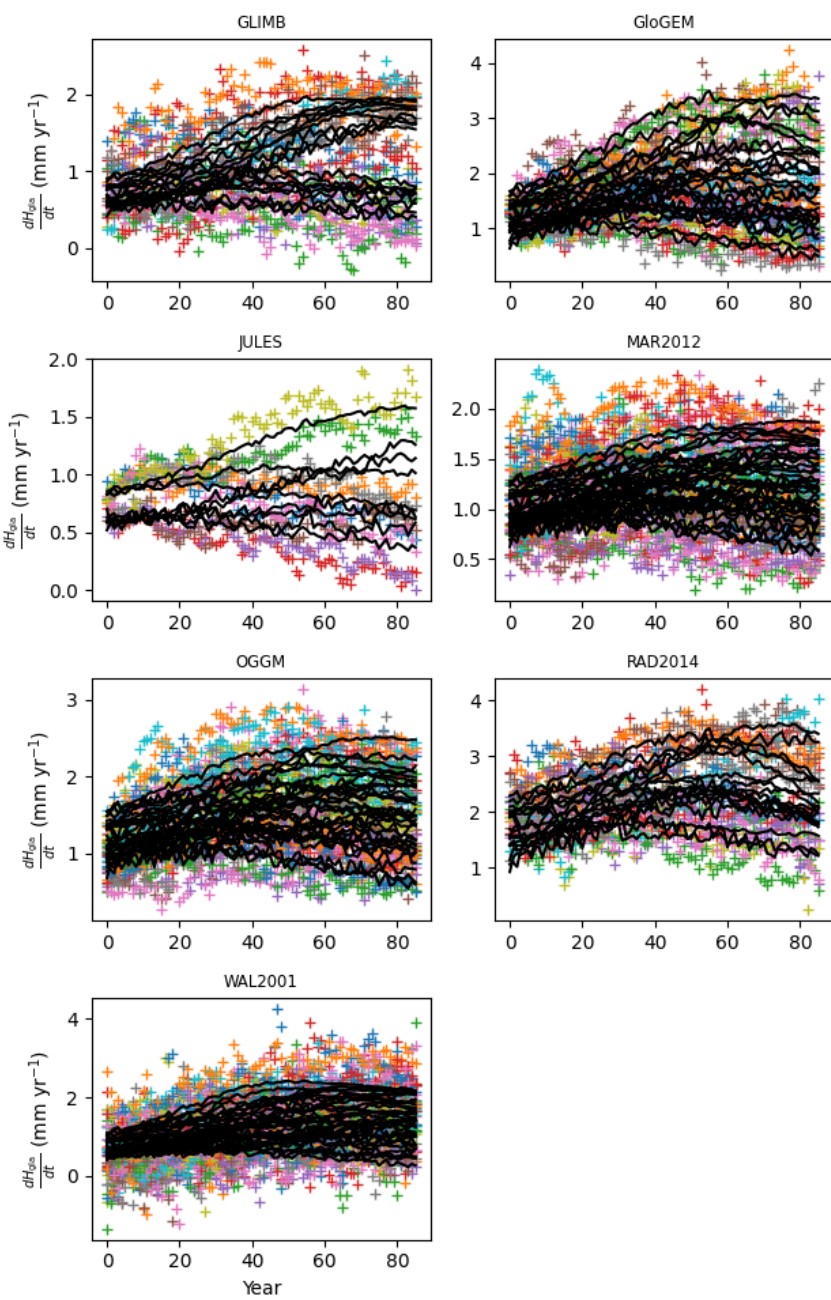

**Figure A7.** Calibration to estimate the prior of Glaciers SLR module parameters ($\Gamma_{gla1}$, $\Gamma_{gla3}$, $\gamma_{gla}$ and $\lambda_{gla0}$). We fit our equation on the compiled outputs from Edwards et al. (2021) for which there is more than one scenario available. Each panel's title displays the name of the model or study used. Coloured markers are the models data while the solid black lines show the fit from Pathfinder.





**Figure A8.** Calibration to estimate the prior of AIS SLR module parameters ($\Lambda_{\mathrm{ais,smb}}$, $\alpha_{\mathrm{ais}}$ and $\lambda_{\mathrm{ais0}}$). We fit our equation on the compiled outputs from Edwards et al. (2021) for which there is more than one scenario available. Each panel's title displays the name of the institute, model and configuration used. Coloured markers are the models data while the solid black lines show the fit from Pathfinder.

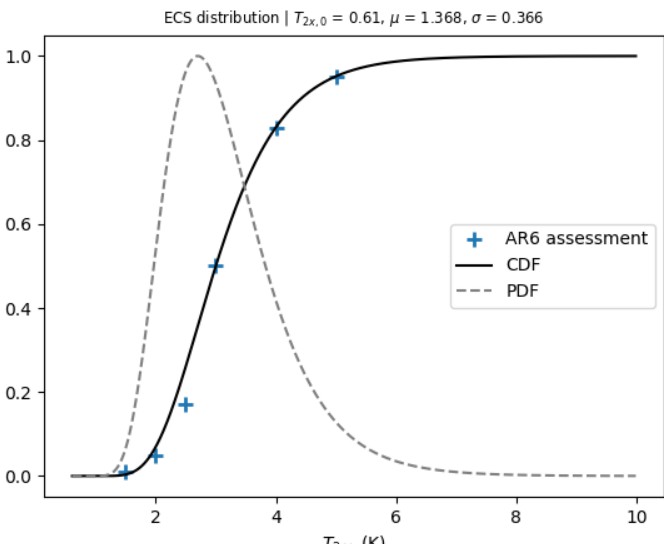

**Figure A9.** Distribution of the logit of the ECS ($T_{2\times}$) inferred from AR6. Blue points are the assessments from AR6, the plain line is the CDF fitted on those assessment and the dashed line is the PDF associated with the CDF (arbitrary scale). The value of the fitted parameters is given above the plot.





| In manual | In code | Description | Units | |
|---|---|---|---|---|
| $R_c$ | RFco2 | CO2 (effective) radiative forcing | W m$^{-2}$ | |
| $R_x$ | ERFx | Non-CO2 effective radiative forcing | W m$^{-2}$ | |
| $R$ | ERF | Effective radiative forcing | W m$^{-2}$ | |
| $T$ | T | Global surface temperature anomaly | K | |
| $T_d$ | Td | Deep ocean temperature anomaly | K | |
| $\mathrm{logit}(\mathrm{ff})$ | logit_ff | Logit of the climate feedback factor (for calib.) | 1 | |
| | | | | |
| $U_{\mathrm{ohc}}$ | OHC | Ocean heat content (anomaly) | W yr m$^{-2}$ | |
| $H_{\mathrm{thx}}$ | Hthx | Thermosteric sea level rise | mm | |
| $H_{\mathrm{gla}}$ | Hgla | Glaciers' contribution to sea level rise | mm | |
| $H_{\mathrm{gis}}$ | Hgis | Grenland ice sheet's contribution to sea level rise | mm | |
| $H_{\mathrm{ais,smb}}$ | Hais_smb | Surface mass balance component of Hais | mm | |
| $H_{\mathrm{ais}}$ | Hais | Antartica ice sheet's contribution to sea level rise | mm | |
| $H_{\mathrm{tot}}$ | Htot | Total sea level rise | mm | |
| $H_{\mathrm{lia}}$ | Hlia | Sea level rise from relaxation after LIA between 1900 and 2005 (for calib.) | mm | |
| | | | | |
| $C_{o,j}$ | Co_j | Change in surface ocean carbon subpools | PgC | $j \in [\![1,5]\!]$ |
| $C_o$ | Co | Change in surface ocean carbon pool | PgC | |
| $C_d$ | Cd | Change in deep ocean carbon pool | | |
| $c_{\mathrm{dic}}$ | dic | Change in surface DIC | $\mu$mol kg$^{-1}$ | |
| $p_{\mathrm{dic}}$ | pdic | Subcomponent of pCO2 | ppm | |
| $p_{\mathrm{CO2}}$ | pCO2 | CO2 partial pressure at the ocean surface | ppm | |
| $F_{\mathrm{ocean}}$ | Focean | Ocean carbon sink | PgC yr$^{-1}$ | |

**Table A1.** Summary of PathFinder's equation variables in climate, sea level and ocean carbon modules. Prog? is for prognostic variables





| In manual | In code | Description | Units |
|---|---|---|---|
| $r_{\mathrm{npp}}$ | r_npp | Relative change in NPP | 1 |
| $r_{\mathrm{fire}}$ | r_fire | Relative change in wildfire intensity | 1 |
| $r_{\mathrm{rh}}$ | r_rh | Relative change in heterotrophic respiration rate | 1 |
| $F_{\mathrm{npp}}$ | NPP | Net primary productivity | PgC yr$^{-1}$ |
| $E_{\mathrm{fire}}$ | Efire | Emissions from wildfire | PgC yr$^{-1}$ |
| $E_{\mathrm{harv}}$ | Eharv | Emissions from harvest and grazing | PgC yr$^{-1}$ |
| $F_{\mathrm{mort}}$ | Fmort | Mortality flux | PgC yr$^{-1}$ |
| $E_{\mathrm{rh1}}$ | RH1 | Litter heterotrophic respiration | PgC yr$^{-1}$ |
| $F_{\mathrm{stab}}$ | Fstab | Stabilization flux | PgC yr$^{-1}$ |
| $E_{\mathrm{rh2}}$ | RH2 | Active soil heterotrophic respiration | PgC yr$^{-1}$ |
| $F_{\mathrm{pass}}$ | Fpass | Passivization flux | PgC yr$^{-1}$ |
| $E_{\mathrm{rh3}}$ | RH3 | Passive soil heterotrophic respiration | PgC yr$^{-1}$ |
| $F_{\mathrm{land}}$ | Fland | Land carbon sink | PgC yr$^{-1}$ |
| $E_{\mathrm{rh}}$ | RH | Heterotrophic respiration | PgC yr$^{-1}$ |
| $C_v$ | Cv | Vegetation carbon pool | PgC |
| $C_{s1}$ | Cs1 | Litter carbon pool | PgC |
| $C_{s2}$ | Cs2 | Active soil carbon pool | PgC |
| $C_{s3}$ | Cs3 | Passive soil carbon pool | PgC |
| $C_s$ | Cs | Total soil carbon pool | PgC |
| | | | |
| $r_{\mathrm{rt}}$ | r_rt | Relative change in permafrost respiration rate | 1 |
| $\bar{a}$ | abar | Theoretical thawed fraction | 1 |
| $a$ | a | Actual thawed fraction | 1 |
| $E_{\mathrm{pf}}$ | Epf | Emissions from permafrost | PgC yr$^{-1}$ |
| $C_{\mathrm{th},j}$ | Cth_j | Thawed permafrost carbon subpools | PgC | $j \in [\![1,3]\!]$ |
| $C_{\mathrm{fr}}$ | Cfr | Frozen permafrost carbon pool | PgC |
| | | | |
| $E_{\mathrm{CO2}}$ | Eco2 | Anthropogenic CO2 emissions | PgC yr$^{-1}$ |
| $C$ | CO2 | Atmospheric CO2 concentration | ppm |
| pH | pH | Surface ocean pH | 1 |

**Table A2.** Summary of PathFinder's equation variables for land carbon, permafrost and atmospheric modules.





| In manual | In code | Description | Units |
|---|---|---|---|
| $\phi$ | phi | Radiative parameter of CO2 | W m$^{-2}$ |
| $T_{2\times}$ | T2x | Equilibrium climate sensitivity | K |
| $\Theta_s$ | THs | Heat capacity of the surface | W yr m$^{-2}$ K$^{-1}$ |
| $\Theta_d$ | THd | Heat capacity of the deep ocean | W yr m$^{-2}$ K$^{-1}$ |
| $\theta$ | th | Heat exchange coefficient | W m$^{-2}$ K$^{-1}$ |
| $\epsilon_{\text{heat}}$ | eheat | Deep ocean heat uptake efficacy | 1 |
| $T_{2\times}^*$ | T2x0 | Minimal value of the ECS distribution (for calib.) | K |
| | | | |
| $\alpha_{\text{ohc}}$ | aOHC | Fraction of energy warming the ocean | 1 |
| $\Lambda_{\text{thx}}$ | Lthx | Proportionality factor of thermosteric SLR | mm m$^2$ W$^{-1}$ yr$^{-1}$ |
| $\lambda_{\text{gla}}$ | lgla0 | Initial imbalance in SLR from Glaciers | mm yr$^{-1}$ |
| $\Lambda_{\text{gla}}$ | Lgla | Maximum contribution to SLR from Glaciers | mm |
| $\Gamma_{\text{gla1}}$ | Ggla1 | Linear sensitivity of steady-state Glaciers SLR to climate | K$^{-1}$ |
| $\Gamma_{\text{gla3}}$ | Ggla3 | Cubic sensitivity of steady-state Glaciers SLR to climate | K$^{-3}$ |
| $\tau_{\text{gla}}$ | tgla | Timescale of Glaciers' contribution to SLR | yr |
| $\gamma_{\text{gla}}$ | ggla | Sensitivity of Glaciers' timescale to climate | K$^{-1}$ |
| $\lambda_{\text{gis}}$ | lgis0 | Initial imbalance in SLR from GIS | mm yr$^{-1}$ |
| $\Lambda_{\text{gis1}}$ | Lgis1 | Linear sensitivity of steady-state GIS SLR to climate | mm K$^{-1}$ |
| $\Lambda_{\text{gis3}}$ | Lgis3 | Cubic sensitivity of steady-state GIS SLR to climate | mm K$^{-3}$ |
| $\tau_{\text{gis}}$ | tgis | Timescale of GIS contribution to SLR | yr |
| $\Lambda_{\text{ais,smb}}$ | Lais_smb | Sensitivity of AIS SMB increase due to climate | mm yr$^{-1}$ K$^{-1}$ |
| $\lambda_{\text{ais}}$ | lais | Initial imbalance in SLR from AIS | mm yr$^{-1}$ |
| $\Lambda_{\text{ais}}$ | Lais | Sensitivity of steady-state AIS SLR to climate | mm K$^{-1}$ |
| $\tau_{\text{ais}}$ | tais | Timescale of AIS contribution to SLR | yr |
| $\alpha_{\text{ais}}$ | aais | Sensitivity of AIS timescale to AIS SLR | mm$^{-1}$ |
| | | | |
| $\alpha_{\text{dic}}$ | adic | Conversion factor for DIC | $\mu$mol kg$^{-1}$ PgC$^{-1}$ |
| $\beta_{\text{dic}}$ | bdic | Inverse-scaling factor for DIC | 1 |
| $\gamma_{\text{dic}}$ | gdic | Sensitivity of pCO2 to climate | K$^{-1}$ |
| $T_o$ | To | Preindustrial surface ocean temperature | °C |
| $\nu_{\text{gx}}$ | vgx | Surface ocean gas exchange rate | yr$^{-1}$ |
| $\gamma_{\text{gx}}$ | ggx | Sensitivity of gas exchange to climate | K$^{-1}$ |
| $\alpha_{o,j}$ | aoc_j | Surface ocean subpools fractions | 1 $j \in [\![1,5]\!]$ |
| $\tau_{o,j}$ | toc_j | Timescales of surface ocean subpools | yr $j \in [\![1,5]\!]$ |
| $\kappa_{\tau_o}$ | k_toc | Scaling factor for timescales of surface ocean subpools | 1 |

**Table A3.** Parameters used in the climate, ocean carbon and sea level modules.



| In manual | In code | Description | Units |
|---|---|---|---|
| $\beta_{\mathrm{npp}}$ | bnpp | Sensitivity of NPP to CO2 (= fertilization effect) | 1 |
| $\alpha_{\mathrm{npp}}$ | anpp | Shape parameter for fertilization effect | 1 |
| $\gamma_{\mathrm{npp}}$ | gnpp | Sensitivity of NPP to climate | $K^{-1}$ |
| $\beta_{\mathrm{fire}}$ | bfire | Sensitivity of wildfire intensity to CO2 | 1 |
| $\gamma_{\mathrm{fire}}$ | gfire | Sensitivity of wildfire intensity to climate | $K^{-1}$ |
| $\beta_{\mathrm{rh}}$ | brh | Sensitivity of heterotrophic respiration to fresh organic matter | 1 |
| $\gamma_{\mathrm{rh}}$ | grh | Sensitivity of heterotrophic respiration to climate | $K^{-1}$ |
| $F_{\mathrm{npp},0}$ | npp0 | Preindustrial NPP | PgC yr$^{-1}$ |
| $\nu_{\mathrm{fire}}$ | vfire | Wildfire intensity | yr$^{-1}$ |
| $\nu_{\mathrm{harv}}$ | vharv | Harvest and grazing rate | yr$^{-1}$ |
| $\nu_{\mathrm{mort}}$ | vmort | Mortality rate | yr$^{-1}$ |
| $\nu_{\mathrm{stab}}$ | vstab | Stabilization rate | yr$^{-1}$ |
| $\nu_{\mathrm{rh1}}$ | vrh1 | Litter heterotrophic respiration rate | yr$^{-1}$ |
| $\nu_{\mathrm{rh23}}$ | vrh23 | Soil (active and passive) respiration rate | yr$^{-1}$ |
| $\nu_{\mathrm{rh3}}$ | vrh3 | Passive soil respiration rate | yr$^{-1}$ |
| $\alpha_{\mathrm{pass}}$ | apass | Fraction of passive soil | 1 |
| $\alpha_{\mathrm{lst}}$ | aLST | Climate scaling factor over permafrost regions | 1 |
| $\gamma_{\mathrm{rt1}}$ | grt1 | Sensitivity of (boreal) heterotrophic respiration to climate | $K^{-1}$ |
| $\gamma_{\mathrm{rt2}}$ | grt2 | Sensitivity of (boreal) heterotrophic respiration to climate (quadratic) | $K^{-2}$ |
| $\kappa_{\mathrm{rt}}$ | krt | Scaling factor for sensitivity of permafrost respiration to climate | 1 |
| $a_{\mathrm{min}}$ | amin | Minimal thawed fraction | 1 |
| $\kappa_{a}$ | ka | Shape parameter for theoretical thawed fraction | 1 |
| $\gamma_{a}$ | ga | Sensitivity of theoretical thawed fraction to climate | $K^{-1}$ |
| $\nu_{\mathrm{thaw}}$ | vthaw | Thawing rate | yr$^{-1}$ |
| $\nu_{\mathrm{froz}}$ | vfroz | Freezing rate | yr$^{-1}$ |
| $\alpha_{\mathrm{th},j}$ | ath_j | Thawed permafrost carbon subpools fractions | 1 $j \in [\![1,3]\!]$ |
| $\tau_{\mathrm{th},j}$ | tth_j | Timescales of thawed permafrost carbon subpools | yr $j \in [\![1,3]\!]$ |
| $\kappa_{\tau_{\mathrm{th}}}$ | k_tth | Scaling factor for timescales of surface ocean subpools | 1 |
| $C_{\mathrm{fr},0}$ | Cfr0 | Preindustrial frozen permafrost carbon pool | PgC |
| $\alpha_{C}$ | aCO2 | Conversion factor for atmospheric CO2 | PgC ppm$^{-1}$ |
| $C_{\mathrm{pi}}$ | CO2pi | Preindustrial CO2 concentration | ppm |
| $\kappa_{\mathrm{pH}}$ | k_pH | Scaling factor for surface ocean pH | 1 |
| $\tilde{\sigma}_{C}$ | std_CO2 | Relative standard deviation of the historical CO2 time series (for calib.) | 1 |
| $\epsilon_{C}$ | ampl_CO2 | Noise amplitude of the historical CO2 time series (for calib.) | ppm |
| $\rho_{C}$ | corr_CO2 | Autocorrelation of the historical CO2 time series (for calib.) | 1 |
| $\tilde{\sigma}_{T}$ | std_T | Relative standard deviation of the historical T time series (for calib.) | 1 |
| $\epsilon_{T}$ | ampl_T | Noise amplitude of the historical T time series (for calib.) | K |
| $\rho_{T}$ | corr_T | Autocorrelation of the historical T time series (for calib.) | 1 |

**Table A4.** Parameters used in the permafrost, land carbon modules and for calibration.





| Parameters | Prior | Unit |
|---|---|---|
| Lgla | 380 | mm |
| Lais | 1200 | mm |
| tgla | 190 | yr |
| tgis | 481 | yr |
| tais | 2093 | yr |
| T2x0 | 0.61 | K |
| adic | 4.49 | umol kg$^{-1}$ PgC$^{-1}$ |
| aoc_1 | 0.87 | 1 |
| aoc_2 | 0.06 | 1 |
| aoc_3 | 0.04 | 1 |
| aoc_4 | 0.02 | 1 |
| aoc_5 | 0.01 | 1 |
| toc_1 | 1.3 | yr |
| toc_2 | 16.7 | yr |
| toc_3 | 65 | yr |
| toc_4 | 348 | yr |
| toc_5 | $10^9$ | yr |
| vrh3 | $8.27 \ 10^{-5}$ | yr$^{-1}$ |
| aLST | 1.87 | 1 |
| grt1 | 0.12 | K$^{-1}$ |
| grt2 | 0.0029 | K$^{-2}$ |
| krt | 1.34 | 1 |
| amin | 0.98 | 1 |
| vthaw | 0.14 | yr$^{-1}$ |
| vfroz | 0.011 | yr$^{-1}$ |
| ath_1 | 0.05 | 1 |
| ath_2 | 0.12 | 1 |
| ath_3 | 0.83 | 1 |
| tth_1 | 18.2 | yr |
| tth_2 | 252 | yr |
| tth_3 | 3490 | yr |
| aCO2 | 2.12 | PgC ppm$^{-1}$ |
| k_pH | 1 | 1 |

**Table A5.** Structural parameters values. Parameters are noted under their code notation, and Tables A3 and A4 provide the corresponding notation in text.





| Parameters | Prior | Posterior | Unit | Parameters | Prior | Posterior | Unit |
|---|---|---|---|---|---|---|---|
| phi | $5.35 \pm 0.54$ | $5.29 \pm 0.54$ | W m$^{-2}$ | T2x | $4.13 \pm 1.37$ | $3.37 \pm 0.77$ | K |
| THs | $8.14 \pm 0.99$ | $8.21 \pm 1.06$ | W yr m$^{-2}$ K$^{-1}$ | THd | $108.6 \pm 61.8$ | $123.8 \pm 57.8$ | W yr m$^{-2}$ K$^{-1}$ |
| th | $0.61 \pm 0.13$ | $0.67 \pm 0.12$ | W m$^{-2}$ K$^{-1}$ | eheat | $1.35 \pm 0.40$ | $1.41 \pm 0.43$ | 1 |
| aOHC | $0.91 \pm 0.02$ | $0.91 \pm 0.02$ | 1 | Lthx | $1.82 \pm 0.21$ | $1.85 \pm 0.23$ | mm m$^2$ W$^{-1}$ yr$^{-1}$ |
| lgla0 | $0.59 \pm 0.24$ | $0.40 \pm 0.21$ | mm yr$^{-1}$ | Ggla1 | $0.34 \pm 0.18$ | $0.34 \pm 0.05$ | mm K$^{-1}$ |
| Ggla3 | $0.022 \pm 0.013$ | $0.022 \pm 0.013$ | mm K$^{-3}$ | ggla | $0.12 \pm 0.09$ | $0.11 \pm 0.07$ | K$^{-1}$ |
| Lgis1 | $82 \pm 45$ | $189 \pm 55$ | mm K$^{-1}$ | Lgis3 | $5.7 \pm 1.4$ | $5.8 \pm 1.5$ | mm K$^{-3}$ |
| Lais_smb | $0.61 \pm 0.19$ | $0.40 \pm 0.10$ | mm K$^{-1}$ yr$^{-1}$ | lais0 | $0.00 \pm 0.11$ | $0.07 \pm 0.08$ | mm yr$^{-1}$ |
| aais | $0.002 \pm 0.003$ | $0.004 \pm 0.003$ | mm$^{-1}$ | lgis0 | $0.33 \pm 0.14$ | $0.35 \pm 0.14$ | mm yr$^{-1}$ |
| k_toc | $1.00 \pm 0.20$ | $0.91 \pm 0.18$ | 1 | vgx | $0.19 \pm 0.06$ | $0.20 \pm 0.07$ | PgC ppm$^{-1}$ yr$^{-1}$ |
| ggx | $0.018 \pm 0.029$ | $0.019 \pm 0.033$ | K$^{-1}$ | To | $18.0 \pm 0.5$ | $18.0 \pm 0.5$ | K |
| bdic | $0.87 \pm 0.08$ | $0.90 \pm 0.09$ | 1 | gdic | $0.04 \pm 0.02$ | $0.04 \pm 0.02$ | K$^{-1}$ |
| npp0 | $48.2 \pm 5.1$ | $46.5 \pm 3.3$ | PgC yr$^{-1}$ | vfire | $0.006 \pm 0.003$ | $0.006 \pm 0.002$ | yr$^{-1}$ |
| vharv | $0.003 \pm 0.003$ | $0.003 \pm 0.002$ | yr$^{-1}$ | vmort | $0.11 \pm 0.01$ | $0.11 \pm 0.01$ | yr$^{-1}$ |
| vstab | $0.32 \pm 0.28$ | $0.30 \pm 0.22$ | yr$^{-1}$ | vrh1 | $0.33 \pm 0.29$ | $0.27 \pm 0.20$ | yr$^{-1}$ |
| vrh23 | $0.024 \pm 0.009$ | $0.024 \pm 0.008$ | yr$^{-1}$ | bnpp | $0.93 \pm 0.37$ | $1.09 \pm 0.25$ | 1 |
| anpp | $0.48 \pm 0.57$ | $0.36 \pm 0.38$ | 1 | gnpp | $-0.014 \pm 0.023$ | $-0.005 \pm 0.024$ | K$^{-1}$ |
| bfire | $-0.05 \pm 0.12$ | $-0.06 \pm 0.14$ | 1 | gfire | $0.052 \pm 0.072$ | $0.044 \pm 0.088$ | K$^{-1}$ |
| brh | $1.06 \pm 0.43$ | $1.01 \pm 0.41$ | 1 | grh | $0.056 \pm 0.053$ | $0.042 \pm 0.035$ | K$^{-1}$ |
| apass | $0.69 \pm 0.19$ | $0.63 \pm 0.20$ | 1 | ga | $0.13 \pm 0.04$ | $0.13 \pm 0.04$ | K$^{-1}$ |
| ka | $2.6 \pm 2.0$ | $2.4 \pm 1.8$ | 1 | k_tth | $0.96 \pm 0.93$ | $1.00 \pm 0.87$ | 1 |
| Cfr0 | $546 \pm 120$ | $538 \pm 122$ | PgC | CO2pi | $278 \pm 3$ | $279 \pm 3$ | ppm |

**Table A6.** Calibrated parameters values before and after Bayesian calibration. Parameters are noted under their code notation, and Tables A3 and A4 provide the corresponding notation in text. The uncertainty correspond to the 1 $\sigma$ uncertainty range.