# Peer review of "Pathfinder v1.0: a Bayesian-inferred simple carbon-climate model to explore climate change scenarios"

_EGUsphere, 2022_

## Author Response (AR1)

**Referee #1**

**Comment #1.1 :**

major comments:

(*) Aim: What is the model aiming to simulate? It is not clear. It would help to spell this out clearly. The model equation have a lot of details, like sub-reservoirs or subpools, but it is unclear if it is the aim to simulate the values of these or are they just a means to simulate something else.

**Response #1.1:**

We assumed that the title (*carbon-climate model*) and e.g. the second sentence of the abstract would make it clear that Pathfinder aims to describe the climate and carbon cycle systems. In addition, we further detailed three requirements we want the model to fulfil while representing simply and accurately the carbon-climate system:

*"Pathfinder was designed to fulfil three key requirements: 1. the capacity to be calibrated using Bayesian inference, 2. the capacity to be coupled with integrated assessment models (IAMs), and 3. the capacity to explore a very large number of climate scenarios to narrow down those compatible with limiting climate impacts."*

Nevertheless, we propose to add the following sentence in introduction: *"Simple climate models (SCMs) typically simulate global mean temperature change caused by either atmospheric concentration changes or anthropogenic emissions, of CO2 and other climatically active species."*; and later on to mention again that Pathfinder is *"CO2-only"*.

**Comment #1.2 :**

(*) Level of complexity: The level of complexity in many of the model equations seems overdone, considering the simplicity of the outcomes of this model. The authors need to better explain why they use highly complex models to achieve fairly simplistic outcomes. Is the level of complexity really needed?

**Response #1.2:**

We built the model using existing formulations, as explained in the second sentence of the abstract, with the goal to keep it as simple and accurate as possible. We propose to add another sentence in the last paragraph of introduction to repeat this information:

*"The model is essentially a integration of existing formulations, adapted to our modelling framework."*

The trade-off between simplicity and complexity/accuracy is already mentioned in the third paragraph of the introduction. While we tend to agree with the referee that the model might be slightly over-complex, we consider it is simply a property of the current version inherited from the formulations compiled to create the model. We however propose to add

an item in the concluding discussion about this aspect, mentioning that future development should strive to reduce complexity wherever possible:

*"First, some parts of the model may well lean too much on the complexity side of the simplicity--accuracy balance we aimed to strike, owing to the creation process of Pathfinder that mostly compiled existing formulations. Future development should therefore strive to reduce complexity wherever possible. The ocean carbon sub-pools and perhaps the land carbon pools, are potential leads in this respect."*

**Comment #1.3 :**

For instance, the ocean carbon model has five subpools. It seems unlikely that this is needed. So why is this done?

Similarly for the land carbon model.

**Response #1.3:**

As indicated on line 114: *"To calculate the ocean carbon sink, we use the mixed-layer impulse response function model from Joos et al. (1996), updated to the equivalent box-model formulation of Strassmann and Joos (2018)."* This is the simplest accurate formulation we could find in the literature.

For the land carbon model, we indicate on line 152 that: *"The land carbon module of Pathfinder is a simplified version of the one in OSCAR (Gasser et al., 2017, 2020)."*

If the referee can suggest simpler but still suitable formulations for these parts of the model, we would be happy to consider them in a future version.

**Comment #1.4 :**

(*) Clarity: Much of the manuscript is really hard to follow. The authors assume a lot of detailed knowledge that only very few readers will have. The manuscript need to be understood to the largest part by a wider community, and I cant see how this manuscript does that. For instance, discussion of figs. 5, 6 and 8 points out details in the text, that I assume can be seen in one of the panels, but it is largely unclear which panel they are talking about, as the text does not refer to the figure nor does it use the same names or acronyms.

**Response #1.4:**

This is a detailed description paper of a numerical implementation (made of about 5000 lines of code) of a mathematical model of the carbon-climate system. It is inherently difficult to understand for those who do not have the prerequisite knowledge on some key aspects

such as the physical processes represented, the concept and mathematics of Bayesian inference, or concrete implementation issues like differential systems solving schemes.

While we agree that, in principle, it would be great if the manuscript could be "understood to the largest part by a wider community", we disagree that the burden of effort should fall exclusively on us. Our paper is not meant to be a book to teach simple climate models: it is merely a description of one such model, in a scientific context that goes much beyond its scope.

Therefore, we do not expect anyone to just pick the model up, do a cursory read of its description paper, and instantly understand and appreciate our work. This can only take time and effort on the reader's side. In other words, this paper is more destined to be read by PhD students rather than by busy professors.

That being said, the referee raises a number of fair points that we can address to further softened the experience of reading such a paper. Notably, we propose to better reference figures in the text, although not every time we refer to them because this would mean ending every single sentence of some sections with "(see Figure X)". We also propose to generate new figures that use the mathematical notations of the equations instead of the original code notations. Finally, we propose to restructure the section of Bayesian calibration for a more logical articulation between the parts about the prior and posterior distribution parameters:

*3 Bayesian calibration*

*3.1 Principle*

*3.2 Implementation*

*3.3 Constraints*

*3.4 Parameters (prior distributions)*

*3.5 Results (posterior distributions)*

*4 Model diagnosis*

*4.1 Historical period*

*4.2 Idealized simulations*

*4.3 Scenarios*

**Comment #1.5 :**

**(*) Parameter fit: The authors use a "Bayesian" approach to fit uncertain model parameters. This approach is hard to follow. If all the parameters are optimised, does this imply some kind of cost function is minimised? If so by how much has the cost function been minimised? How is the success of the optimisation measured?**

**Response #1.5:**

A Bayesian calibration does not optimise parameters in the usual sense, for instance in terms of a distance to target observations. It calculates the probability distributions of the parameters, given knowledge of the probabilistic distributions of a number of outcomes of the model (the constraints, i.e. the observations). This is in theory simply done using Bayes' formula.

However, in all but trivial cases, the actual posterior distributions (i.e. the result of Bayes formula) is not tractable and cannot be expressed using closed-form formula. Therefore, one must use numerical methods to approximate these distributions.

Here, because the system to be optimised is of significant size, we use a specific variational inference algorithm that approximates the sought posterior distributions by minimizing a quantity known as the evidence lower bound (ELBO). This whole sampling process is done in 100,000 iterations, and we checked convergence of the ELBO. Further details on Bayesian inference and the variational inference algorithm are available in two papers we cite: Kucukelbir et al. (2017) and Beir et al., (2017; doi:10.1080/01621459.2017.1285773).

We propose to add an additional sentence in the section 3.1 on the principle of Bayesian calibration, to better explain the concept and role:

*"Summarily, the Bayesian calibration updates the joint distribution of parameters to make it as compatible with the constraints as possible given their prior estimates, which increases internal coherence of Pathfinder by excluding combination of parameters that are unlikely."*

Similarly, we propose to rephrase and expand parts of the implementation subsection to make it a new paragraph:

*"The Bayesian procedure itself is implemented using the Python computer language, and specifically the PyMC3 package (Salvatier et al., 2016). The solving of equation 47 and its normalization are done using the package's full-rank Automatic Differentiation Variational Inference (ADVI) algorithm (Kucukelbir et al., 2017), with 100,000 iterations (and default algorithm options). The choice of variational inference instead of Markov chain Monte Carlo is motivated by the significant size our model (Beir et al., 2017) and the speed of ADVI. An additional strength of the full-rank version of the ADVI algorithm is its ability to generate correlated posterior distributions even if the prior ones are uncorrelated. Convergence of the algorithm was controlled through convergence of the ELBO metric (Kucukelbir et al., 2017). All results presented hereafter are obtained through drawing 2000 sets of parameters – that we call configurations – from the posterior or prior distributions."*

**Comment #1.6 :**

(*) Discussion of Figures: The figure are not properly or not at all (Fig.4?) discussed. The text discussion seem to discuss the figures in some cases, but it is unclear what figure or panel is discussed (e.g. Fig. 5,6 and 8).

**Response #1.6:**

Figure 4 was not properly discussed. We propose to add the following in section 2.6 as follows: *Figure 4 gives a representation of the permafrost module as described in the following.*

We acknowledge that figures are not directly cited every time they are discussed as an entire section (section 3.1) is dedicated to the discussion of figures 5 and 6. Hence, we wrote as an introduction to that section: "*Figure 5 shows the prior and posterior distributions of the model's parameters, while Figure 6 shows those of the constraints.*"

**Comment #1.7:**

line 114 "... classic ...": Remove "classic".

**Response #1.7:**

Done.

**Comment #1.8:**

line 116, five boxes: Why is this done? Why is this needed?

**Response #1.8:**

As indicated on line 114: "*To calculate the ocean carbon sink, we use the classic mixed-layer impulse response function model from Joos et al. (1996), updated to the equivalent box-model formulation of Strassmann and Joos (2018).*" This is the simplest formulation we could find in the literature. As this paper directly uses an existing formulation, we invite the referee to check Strassmann and Joos (2018) for more details.

**Comment #1.9:**

eq. [17]: What is $C_p$?

**Response #1.9:**

As indicated on line 65 after eq. [4] or in Table A4, $C_{pi}$ is the preindustrial atmospheric $CO_2$ concentration.

**Comment #1.10:**

eq. [18]: What is C?

**Response #1.10:**

As indicated on line 41 or in Table A2, C is the global atmospheric CO2 concentration.

**Comment #1.11:**

eq. [21]: Why is this not a function of C_o but C?

**Response #1.11:**

As indicated on lines 144-147: "*While in the real world, ocean acidification is directly related to the carbonate chemistry and the ocean uptake of anthropogenic carbon, we do not have a simple formulation at our disposal that could link it to our ocean carbon cycle module. We therefore use a readily available emulation of the surface ocean acidification (pH) that links it directly to the atmospheric concentration of CO2 (Bernie et al., 2010).*"

We also noted one line after: "*We note that this approach is reasonable for the surface ocean, as it quickly equilibrates with the atmosphere (but it would not work for the deep ocean).*"

We propose to add mention of this limitation in the concluding remark that pertain to the ocean carbon cycle module: "*and properly connected to ocean pH and the ocean of the climate module*".

**Comment #1.12:**

r_npp, eq. [24]: The equation is not easy to understand. It may help to put this into words a little bit. How does r_npp change if C or T goes up?

**Response #1.12:**

Noting that we already explained the role of the alpha_npp shape parameter (as it tends towards 0, the whole function tends towards a logarithm), we propose to add the following information regarding the effect of C: "*whereby NPP increases with atmospheric CO2*"; and the following regarding T: "*that can be positive or negative*".

**Comment #1.13:**

r_fire eq. [28]: Why do wildfires (r_fire) depend on CO2 and not just T?

**Response #1.13:**

This equation is, as for all land carbon equations, highly inspired from OSCAR equations (Gasser et al., 2017). In Gasser et al., 2017, they justify that: "*The change in fire intensity is a function of changes in atmospheric CO2 – used as a proxy variable to encompass various effects such as change in leaf area index, which would help wildfires to spread, or change in evapotranspiration and thus in soil moisture that would reduce their intensity – […].*"

We similarly added a parenthesis explaining this factor: "*as a proxy of changes in leaf area index and evapotranspiration*".

**Comment #1.14:**

section 3.3 constraints: It is unclear how the constraints relates to the previous section parameter fits. Does the constraints lead to changes in the parameters of the model? Are the same parameters changed again? It needs a bit more discussion.

**Response #1.14:**

Section 3.2 explains how the prior estimation (before calibration) of the parameters is obtained. Section 3.3 explains the set of observation **x** used in the Bayesian calibration (see Section 3.1). By essence of a Bayesian calibration, the constraints lead to changes in the parameters from their initially assumed value and uncertainty distribution: that is the meaning of the words "prior" and "posterior".

This was already explained by the following sentence of section 3.1: "*The approach consists in deducing joint probability distributions of parameters from a priori knowledge on those distributions and on distributions of observations of some of the model's state variables.*"

Hopefully, the addition sentence we propose in Response #1.5 will makes this even clearer: "*Summarily, the Bayesian calibration updates the joint distribution of parameters to make it as compatible with the constraints as possible given their prior estimates, which increases internal coherence of Pathfinder by excluding combination of parameters that are unlikely.*"

**Comment #1.15:**

section 4.1 Posterior distributions: This section is hard to follow. I failed to understand it, but it is central to the study. It seems the text is discussing the Figs. but it is unclear how. Example: "ECS (T2×) is the parameter with the strongest adjustment ..." Is this something the reader can see in a Figure? Which Figure? Why does the text refer to different names that have no match in the Figures?

**Response #1.15:**

Indeed, this section is discussing the figures, as explained by the introductory sentence of the section: "*Figure 5 shows the prior and posterior distributions of the model's parameters, while Figure 6 shows those of the constraints.*"

We propose to further introduce this section with the following new sentence: "*The following subsections discuss the adjustments between the prior and posterior parameters that are the results of the Bayesian calibration, as well as the matching of the constraints.*"

To take the example of ECS (T2x) raised by the referee, one must simply look at Figure 5, find the panel entitle T2x (which is the second panel), and compare the black and blue distributions to compare the prior and posterior distributions of T2x.

Alternatively, one could look up the values of the prior and posterior parameters in Table A6, although this one was not properly referenced. We propose to correct this by adding the following sentence in introduction of the section: *"Prior and posterior values of the parameters can also be retrieved from in Table A6."*

Finally, regarding the change of notations, this was done because mathematical notations are much harder to read on the figures than the code notations, and the correspondence between the two are provided in Tables A1 to A4. However, we propose to generate new figures that use the mathematical notations.

**Comment #1.16:**

Fig. 5 Distributions before and after the Bayesian calibration: It is unclear what the message of this analysis is. It seems there is not "truth" in this, so what is the reader suppose to see here? Is the before distribution assume to be the "truth"?

**Response #1.16:**

See Response #1.5. There is no truth in Bayesian statistics, just an updated estimate of the parameters' distributions that assimilates the information contained in the constraints. The updated values are assumed better because they contain more information.

**Comment #1.17:**

Fig. 8; temperature driven setup: this needs a bit more discussion. How is this simulation constructed? The upper panels are by construction a perfect fit? Where and how is T and the atmospheric CO2 evaluated? It seems it is not down here as T is prescribed.

**Response #1.17:**

We are unsure what kind of discussion the referee expects, here. As stated in section 4.2, lines 465-466: "*Figure 8 gives the time series from 1900 to 2021 of six key variables. GMST and atmospheric CO2 match very well the historical observations, by construction of these input time series.*" This answers the referee's first question.

Subsection 3.2.8 "*Historical CO2 and GMST*" explains in detail how T and CO2 are determined from historical time series to be used as input to the Bayesian calibration. The output of the Bayesian calibration for these two time series is shown in Figure 8, which answers the second question.

**Comment #1.18:**

model uncertainty ranges in Figs. 8 and 9: Where does the uncertainty ranges for the model come from?

**Response #1.18:**

The uncertainty simply comes from the distribution of parameters, although we failed to clearly state this in the submitted version of the paper. We propose to add the following sentence: "*All results presented hereafter are obtained through drawing 2000 sets of parameters – that we call configurations – from the posterior or prior distributions.*"

**Comment #1.19:**

Fig. 9: What are the columns 2 and 3 showing? Is this the integrated carbon uptake for land and ocean?

**Response #1.19:**

Yes it is. We propose to modify the caption as: "*Time series of GMST change, integrated land carbon uptake and integrated ocean carbon uptake for idealized experiments (abrupt-2xCO2, 1pctCO2, 1pctCO2-bgc and 1pctCO2-rad), and projections according to SSP scenarios in Pathfinder.*"

**Comment #1.20:**

line 515 "The ocean carbon storage appears overestimated by 5% to 20% by Pathfinder across SSP scenarios": relative to what reference?

**Response #1.20:**

We state on lines 507-509 that: "*Table 3 shows a comparison of the projected changes in GMST to the CMIP6 estimates (Lee et al., 2021, Table 4.2), and of carbon pools to Liddicoat et al. (2021) (since this was not directly reported in the AR6).*" Therefore, the ocean carbon storage appears overestimated compared to the ocean carbon storage estimated in Liddicoat et al., 2021.

**Referee #2**

**Comment #2.1:**

1. GENERAL COMMENT

The editors have advised me that this paper has been submitted to GMD as a 'model description' paper. Accordingly, I have assessed it against the GMD criteria for such papers, (in about/manuscript\_types.html on the GMD website)

In the primary text, I find the introduction is short of detail on the context and that the detail about accessing the model is insufficient.

**Response #2.1:**

The detail about accessing the model were indeed insufficient, and we have added those in an update of the model's repository.

Regarding detail on the context, however, we are unsure what the referee expects, exactly. We explain the philosophy behind this type of models, the philosophy behind our own model, and cite a number of papers on the topic. We propose to add a paragraph in the Introduction to better position Pathfinder with respect to other SCMs: "*Compared to other SCMs (Nicholls et al., 2020), Pathfinder is much simpler than models like MAGICC (Meinshausen et al., 2011), OSCAR (Gasser et al., 2017) or even HECTOR (Hartin et al., 2015). It is comparable in complexity to FaIR (Smith et al., 2018) or BernSCM (Strassmann and Joos, 2018), although it is closer to the latter as it trades off an explicit representation of non-CO2 species for one of the carbon cycle's main components. This choice was made to help calibration, keep the model invertible, and be compatible with IAMs such as DICE (Nordhaus et al., 2017). While most SCMs are calibrated using procedures that resemble Bayesian inference (Nicholls et al., 2021), Pathfinder relies on an established algorithm whose implementation is fully tractable, and that allows for an annual update as observations of atmospheric CO2 and global temperature become available.* "*

**Comment #2.2:**

The specifications note that the text should describe the software and hardware requirements for running the model. This is missing.

**Response #2.2:**

Indeed. We propose to add this information in Appendix:

"*Pathfinder has been developed and run in Python (v3.7.6), preferentially using IPython (v7.19.0). Currently, packages required to run it are NumPy (v1.19.2), SciPy (v1.5.2) and Xarray (v0.16.0), and it has hard-coded dependencies on PyMC3 (v3.8) and Theano (v1.0.4) that are in fact used only for calibration. Other versions of Python or these packages were not tested.*

*The calibration procedure takes about 9 hours to run on a desktop computer (with a base speed of 3.4 GHz). Simple use of the model is much faster: the idealized experiments and SSP scenarios for this description paper, which represent 2984 simulated years, were run in about 20 minutes for all 2000 configurations and on a single core. A single simulated year takes a few tenth of a second, although a number of options in the model can drastically alter this performance. Note also that this scales sub-linearly with the amount of configurations or scenarios because of the internal workings of the Xarray package, albeit at the cost of increased demand in random-access memory.*"

We also propose to add in Appendix the changelog and 'known issues' that we added in the updated README file.

**Comment #2.3:**

The specifications say that a publication should consist of text, code and user manual. The authors acknowledge the incomplete nature of the current state of their site, describing it as "messy". I have downloaded and expanded the zip file and extracted files manual.pdf and manual.md. As a user manual, I find it inadequate because it is only a brief download instruction, a list of model variables and a list of equations that presumably repeat those in the text and essentially nothing about how to use it. The latter two sections are faulty (unreadable) as described below. If the authors want people to try the model, they need to do better.

**Response #2.3:**

We are very sorry we did not see this error when exporting the .md files into .pdf. This has been corrected in version 1.0.1 (in both the GitHub and Zenodo repositories). The new Zenodo repository is available here: https://doi.org/10.5281/zenodo.7003848

However, this was merely a displaying error, and most of the requirements were already provided as they should have been: main text as a description of the model, and code fully usable. We acknowledge, however, there was room for improvement in making the readme and manual files more user-friendly, which we did in this small update.

**Comment #2.4:**

My recommendation is that the authors withdraw the paper and resubmit when the online information is better organised. Releasing the model at this stage is doing a disservice to the modelling community and the authors themselves.

**Response #2.4:**

We find that greatly exaggerated. The small display error is now corrected, and additional information including a very simple example of use has been added.

**Comment #2.5:**

2. SPECIFIC COMMENTS

2.1 Without doing a line-by-line check, it seems to me that the description is sufficient to enable someone else to implement equivalent code to integrate the model equations.

The ability to run with multiple choices of two input series is more problematic (without actually writing 6 separate versions (sharing some code) using the principles described by Wigley 1991).

There does not seem to be enough information to independently implement the model calibrations. (This is not necessarily a GMD requirement).

**Response #2.5:**

The ability to run with multiple choices of two input series was indeed implemented through separate versions of the model (we actually implemented only 4 of the 6 possibilities). The code is in the files whose names start with 'mod_' located in the 'core_fct' folder. We propose to add the following detail to the manuscript: "*(in terms of code, implemented as separate versions of the model)*".

Regarding the calibration, it is simply not true: one can execute the 'run_calib_and_hist.py' file located in the 'run_scripts' folder to repeat exactly the calibration. One can also backtrack one's way into the various functions and sub-functions called when executing said script, to further pick the calibration procedure apart.

All this is now easier to figure out, thanks to the clean and expanded manual file.

**Comment #2.6:**

2.2 The manuals and online information.
(This is based on the contents of the zip file from the zemodo repository).
The manual consists of 3 sections:
* Run a simulation
* Notation
* Equations

2.2a. Manual.pdf

Many aspects are not being displayed, even in the list of variables. My guess is that the pdf has been created in a form that makes use of external fonts on the author's system, rather than importing the requisite font descriptions (maybe for math fonts) into the pdf file. The whole of the equation section fails to display.

**Comment #2.6:**

Corrected. Apologies again for this. This was because of failed image imports, as until very recently the GitHub .md interpreter did not support math display.

**Comment #2.7:**

2.2b.  Manual. md. This is the other way round. The equation section displays as math characters, presumably reproducing what is in the text. However the Notation section is just an unformatted, unspaced string of ascii characters with occasional math symbols, presumably those missing from column 1 in the equation section in manual.pdf.

**Comment #2.7:**

Corrected.

**Comment #2.8:**

2.3 The online files. (From the .zip)

As noted, the "manual" does not describe what is needed to run the model, in particular what input files are involved? For example what is the role of the .nc (netcdf?) files and the .xlsx files.
How do these relate to the RCMIP standards noted by Nicholls 2020, 2021? Similarly, the comments at the beginning of each python file give no information about what the file is for and even if is is the main file or a module called by one or more main files.

**Comment #2.8:**

Now provided in the expanded manual file, specifically as a list of folders and/or files and a brief description of the content.

We note here that the file formats we use are not related to the RCMIP standards (that we find inconvenient).

**Comment #2.9:**

2.4 Role of model.

The authors state that they have identified the need for a new model, citing Nicholls 2020.

This is not obvious from that paper. More detail should be given, being more specific about the niche that the model is intended to fill and the relation between this paper and earlier work such as Meinshausen 2009.

**Response #2.9:**

In the introduction, we clearly state the three requirements we want our model to fulfil as none of the existing model fulfil them simultaneously (line 21-25): *"Pathfinder was designed to fulfil three key requirements: 1. the capacity to be calibrated using Bayesian inference, 2. the capacity to be coupled with integrated assessment models (IAMs), and 3. the capacity to explore a very large number of climate scenarios to narrow down those compatible with limiting climate impacts."*

We propose to add a paragraph to better position Pathfinder with respect to other SCMs: "*Compared to other SCMs (Nicholls et al., 2020), Pathfinder is much simpler than models like MAGICC (Meinshausen et al., 2011), OSCAR (Gasser et al., 2017) or even HECTOR (Hartin et al., 2015). It is comparable in complexity to FaIR (Smith et al., 2018) or BernSCM (Strassmann and Joos, 2018), although it is closer to the latter as it trades off an explicit representation of non-CO2 species for one of the carbon cycle's main components. This choice was made to help calibration, keep the model invertible, and be compatible with IAMs such as DICE (Nordhaus et al., 2017). While most SCMs are calibrated using procedures that resemble Bayesian inference (Nicholls et al., 2021), Pathfinder relies on an established algorithm whose implementation is fully tractable, and that allows for an annual update as observations of atmospheric CO2 and global temperature become available.* "

**Comment #2.10:**

2.5 Mixed layer pools.

The other referee raised the issue of the meaning and number of the mixed layer pools. This structure has no physical meaning and simply represents a mathematical transformation to go from the impulse response function of Joos 1996 to a set of differential equations. This is presumably a well-known transformation, although I first became aware of it through Wigley 1991. The number of pools derives from the number of exponentials in the impulse response function. An approach (derived from electrical engineering) for deriving lower order approximations is given in my recent paper Enting 2022.

The structure has the 5 pools operating in parallel. Writing the equations in matrix vector form as dx/dt = A x + b.f, where f is the input, it should be possible to take linear combinations (applied to both rows and columns of A) such that only one element of b is zero, and maybe even end up with A in the tridiagonal form representing pools in series.

**Response #2.10:**

We thank the referee for the reference, and will look into it when we work on revamping and hopefully simplifying the ocean carbon module.

We also note that A is already a diagonal matrix, therefore the pools are solved independently from one another, and we do not see any way to make the calculation faster (except, of course, reducing the number of pools).

**Comment #2.11:**

2.6 Calibration
At one point is is noted that the philosophy that priors are defined by the more complex earth system models, with posterior values determined by applying observations as constraints. If this is true throughout, it should be stated in both the abstract and the introduction. Of course, this sort of Bayesian calibration is vulnerable to the likelihood that such priors will be drawing on some of the same information as the constraint data.

**Response #2.11:**

We propose to add this sentence upfront in the introduction (line 31-32): "*The philosophy of Pathfinder is to use complex models as prior information and only real world observations (or assessment combining many lines of evidence) as constraints for the Bayesian calibration.*"

The referee makes a good point regarding degenerate information. So we also propose to add the following brief discussion in the 'principle' subsection on the Bayesian calibration:

"*Such a Bayesian calibration is vulnerable to the possibility that the priors draw on the same information as the constraints. However, given that Pathfinder is a patchwork of emulators whose parameters are obtained independently from one another and following differing experimental setups, we expect that the coherence of information contained within the priors and the constraints is very low. Our choice of using only complex models as prior information and only observations and assessments as constraints also aims at limiting this vulnerability.*"

**Comment #2.12:**

An extra sentence on the PyMC3 operation would be helpful.

**Response #2.12:**

We propose to significantly expand this section. In addition to the short paragraph of Response #2.11, comments by Referee #1 motivated us to add the following sentences:

"*Summarily, the Bayesian calibration updates the joint distribution of parameters to make it as compatible with the constraints as possible given their prior estimates, which increases internal coherence of Pathfinder by excluding combination of parameters that are unlikely.*"

and

"*The Bayesian procedure itself is implemented using the Python computer language, and specifically the PyMC3 package (Salvatier et al., 2016). The solving of equation 47 and its normalization are done using the package's full-rank Automatic Differentiation Variational Inference (ADVI) algorithm (Kucukelbir et al., 2017), with 100,000 iterations (and default algorithm options). The choice of variational inference instead of Markov chain Monte Carlo is motivated by the significant size our model (Beir et al., 2017) and the speed of ADVI. An additional strength of the full-rank version of the ADVI algorithm is its ability to generate correlated posterior distributions even if the prior ones are uncorrelated. Convergence of the algorithm was controlled through convergence of the ELBO metric (Kucukelbir et al., 2017). All results presented hereafter are obtained through drawing 2000 sets of parameters – that we call configurations – from the posterior or prior distributions.*"

In this section on technical implementation of the Bayesian calibration, we also propose to explicit the IMEX solving scheme we use for the differential system.

**Comment #2.13:**

2.7 Input time series.
It is unclear what happens to the two input time series (which would seem to be needed at all times) as the calibration adjusts the 6 parameters that characterise these series.

**Response #2.13:**

What happens to the input time series during calibration is illustrated in the first two panels of Figure 8. In addition, the changes in the distributions of the 6 parameters (between prior and posterior) are shown in Figure 5, in the last 6 panels.

**Comment #2.14:**

3 TECHNICAL CORRECTIONS

Line 296. Cross reference not working.

**Response #2.14:**

Done.

**Comment #2.15:**

Line 440.  to be  -> from being

**Response #2.15:**

Done.

**Comment #2.16:**

Line 549 stricken -> struck

**Response #2.16:**

Done.

**Comment #2.17:**

Caption for table A1 notes an indicator of prognostic variables. This does not seem to be used.

**Response #2.17:**

Removed.

**Comment #2.18:**

I have not made any attempt to check against the guideline that papers should use either US or UK English, but not mix them.

**Response #2.18:**

This will be checked during copy editing.